# Visual Explanation by Interpretation: Improving Visual Feedback Capabilities of Deep Neural Networks

José Oramas M.[*]            Kaili Wang[*]            Tinne Tuytelaars
*KU Leuven, ESAT-PSI*

## Abstract

Interpretation and explanation of deep models is critical towards wide adoption of systems that rely on them. In this paper, we propose a novel scheme for both interpretation as well as explanation in which, given a pretrained model, we automatically identify internal features relevant for the set of classes considered by the model, without relying on additional annotations. We *interpret* the model through average visualizations of this reduced set of features. Then, at test time, we *explain* the network prediction by accompanying the predicted class label with supporting visualizations derived from the identified features. In addition, we propose a method to address the artifacts introduced by strided operations in deconvNet-based visualizations. Moreover, we introduce an8Flower, a dataset specifically designed for objective quantitative evaluation of methods for visual explanation. Experiments on the MNIST, ILSVRC12, Fashion144k and an8Flower datasets show that our method produces detailed explanations with good coverage of relevant features of the classes of interest.

## 1 Introduction

Methods based on deep neural networks (DNNs) have achieved impressive results for several computer vision tasks, such as image classification, object detection and image generation. Combined with the general tendency in the Computer Vision community of developing methods with a focus on high quantitative performance, this has motivated the wide adoption of DNN-based methods, despite the initial skepticism due to their black-box characteristics. In this work, we aim for more visually-descriptive predictions and propose means to improve the quality of the visual feedback capabilities of DNN-based methods. Our goal is to bridge the gap between methods aiming at model *interpretation*, i.e., understanding what a given trained model has actually learned, and methods aiming at model *explanation*, i.e., justifying the decisions made by a model.

Model interpretation of DNNs is commonly achieved in two ways: either by a) manually inspecting visualizations of every single filter (or a random subset thereof) from every layer of the network (Yosinski et al. (2015); Zeiler & Fergus (2014)) or, more recently, by b) exhaustively comparing the internal activations produced by a given model w.r.t. a dataset with pixel-wise annotations of possibly relevant concepts (Bau et al. (2017); Fong & Vedaldi (2018)). These two paths have provided useful insights into the internal representations learned by DNNs. However, they both have their own weaknesses. For the first case, the manual inspection of filter responses introduces a subjective bias, as was evidenced by Gonzalez-Garcia et al. (2017). In addition, the inspection of every filter from every layer becomes a cognitive-expensive practice for deeper models, which makes it a noisy process. For the second case, as stated by Bau et al. (2017), the interpretation capabilities over the network are limited by the concepts for which annotation is available. Moreover, the cost of adding annotations for new concepts is quite high due to its pixel-wise nature. A third weakness, shared by both cases, is inherited by the way in which they generate spatial filter-wise responses, i.e., either through deconvolution-based heatmaps (Springenberg et al. (2015); Zeiler & Fergus (2014)) or by up-scaling the activation maps at a given layer/filter to the image space (Bau et al. (2017); Zhou et al. (2016)). On the one hand, deconvolution methods are able to produce heatmaps with high level of detail from any filter in

---

*denotes equal contribution

Figure 1: Left: Proposed training/testing pipeline. Center: Visual explanations generated by our method. Predicted class labels are enriched with heatmaps indicating the pixel locations, associated to the features, that contributed to the prediction. Note these features may come from the object itself as well as from the context. On top of each heatmap we indicate the number of the layer where the features come from. The layer type is color-coded (green for convolutional and pink for fully connected). Right: Visualization comparison. Note how our heatmaps attenuate the grid-like artifacts introduced by deconvnet-based methods at lower layers. At the same time, our method is able to produce a more detailed visual feedback than up-scaled activation maps.

the network. However, as can be seen in Fig. 1 (right), they suffer from artifacts introduced by strided operations in the back-propagation process. Up-scaled activation maps, on the other hand, can significantly lose details when displaying the response of filters with large receptive field from deeper layers. Moreover, they have the weakness of only being computable for convolutional layers.

In order to alleviate these issues, we start from the hypothesis proven by Bau et al. (2017); Yosinski et al. (2015), that only a small subset of the internal filters of a network encode features that are important for the task that the network addresses. Based on that assumption, we propose a method which, given a trained DNN model, automatically identifies a set of relevant internal filters whose encoded features serve as indicators for the class of interest to be predicted (Fig. 1 left). These filters can originate from any type of internal layer of the network, i.e., *convolutional, fully connected*, etc. Selecting them is formulated as a $\mu$-*lasso* optimization problem in which a sparse set of filter-wise responses are linearly combined in order to predict the class of interest. At test time, we move from interpretation to explanation. Given an image, a set of identified relevant filters, and a class prediction, we accompany the predicted class label with heatmap visualizations of the top-responding relevant filters for the predicted class, see Fig. 1 (center). In addition, by improving the resampling operations within deconvnet-based methods, we are able to address the artifacts introduced in the back-propagation process, see Fig. 1 (right). The code and models used to generate our visual explanations can be found in the following link [1]. Overall, the proposed method removes the requirement of additional expensive pixel-wise annotation, by relying on the same annotations used to train the initial model. Moreover, by using our own variant of a deconvolution-based method, our method is able to consider the spatial response from any filter at any layer while still providing visually pleasant feedback. This allows our method to reach some level of explanation by interpretation.

Finally, recent approaches to evaluate explanation methods measure the validity of an explanation either via user studies (Zeiler & Fergus (2014); Selvaraju et al. (2017)) or by measuring its effect on a proxy task, e.g. object detection/segmentation (Zhou et al. (2015); Zhang et al. (2016)). While user studies inherently add subjectivity, benchmarking through a proxy task steers the optimization of the explanation method towards such task. Here we propose an objective evaluation via *an8Flower*, a synthetic dataset where the discriminative feature between the classes of interest is controlled. This allows us to produce ground-truth masks for the regions to be highlighted by the explanation. Furthermore, it allows us to quantitatively measure the performance of methods for model explanation.

The main contributions of this work are four-fold. First, we propose an automatic method based on feature selection to identify the network-encoded features that are important for the prediction of a given class. This alleviates the requirement of exhaustive manual inspection or additional expensive pixel-wise annotations required by existing methods. Second, the proposed method is able to provide visual feedback with higher-level of detail over up-scaled raw activation maps and improved quality over recent deconvolution+guided back-propagation methods. Third, the proposed method is general enough to be applied to any type of network, independently of the type of layers that compose it. Fourth, we release a dataset and protocol specifically designed for the evaluation of methods for model explanation. To the best of our knowledge this is the first dataset aimed at such task.

---

[1] http://homes.esat.kuleuven.be/~joramas/projects/visualExplanationByInterpretation

This paper is organized as follows: in Sec. 2 we position our work w.r.t. existing work. Sec. 3 presents the pipeline and inner-workings of the proposed method. In Sec. 4, we conduct a series of experiments evaluating different aspects of the proposed method. We draw conclusions in Sec. 5.

## 2 RELATED WORK

**Interpretation.** Zeiler & Fergus (2014); Zhou et al. (2015) proposed to visualize properties of the function modelled by a network by systematically covering (part of) the input image and measuring the difference of activations. The assumption is that occlusion of important parts of the input will lead to a significant drop in performance. This procedure is applied at test time to identify the regions of the image that are important for classification. However, the resolution of the explanation will depend on the region size. Another group of works focuses on linking internal activations with semantic concepts. Escorcia et al. (2015) proposed a feature selection method in which the neuron activations of a DNN trained with object categories are combined to predict object attributes. Similarly, Bau et al. (2017); Fong & Vedaldi (2018); Zhang et al. (2018) proposed to exhaustively match the activations of every filter from the convolutional layers against a dataset with pixel-wise annotated concepts. While both methods provide important insights on the semantic concepts encoded by the network, they are both limited by the concepts for which annotation is available. Similar to Escorcia et al. (2015), we discover relevant internal filters through a feature selection method. Different from it, we link internal activations directly to the same annotations used to train the initial model. This removes the expensive requirement of additional annotations. A third line of works aims at discovering frequent visual patterns (Doersch et al. (2015); Rematas et al. (2015)) occurring in image collections. These patterns have a high semantic coverage which makes them effective as means for summarization. We adopt the idea of using visualizations of (internal) mid-level elements as means to reveal the relevant features encoded, internally, by a DNN. More precisely, we use the average visualizations used by these works in order to interpret, visually, what the network has actually learned.

**Explanation.** For the sake of brevity, we ignore methods which generate explanations via bounding boxes (Karpathy & Fei-Fei (2017); Oramas M. & Tuytelaars (2016)) or text (Hendricks et al. (2016)), and focus on methods capable of generating visualizations with pixel-level precision. Zeiler & Fergus (2014) proposed a deconvolutional network (Deconvnet) which uses activations from a given top layer and reverses the forward pass to reveal which visual patterns from the input image are responsible for the observed activations. Simonyan et al. (2014) used information from the lower layers and the input image to estimate which image regions are responsible for the activations seen at the top layers. Similarly, Bach et al. (2015) decomposed the classification decision into pixel-wise contributions while preserving the propagated quantities between adjacent layers. Later, Springenberg et al. (2015) extended these works by introducing "guided back-propagation", a technique that removes the effect of units with negative contributions in forward and backward pass. This resulted in sharper heatmap visualizations. Zhou et al. (2016) propose Global Average Pooling, i.e., a weighted sum over the spatial locations of the activations of the filters of the last convolutional layer, which results in a class activation map. Finally, a heatmap is generated by upsampling the class activation map to the size of the input image. Selvaraju et al. (2017) extended this by providing a more efficient way for computing the weights for the activation maps. Recently, Chattopadhyay et al. (2018) extended this with neuron specific weights with the goal of improving object localization on the generated visualizations. Here, we take DeconvNet with guided-backpropagation as starting point given its maturity and ability to produce visual feedback with pixel-level precision. However, we change the internal operations in the backward pass with the goal of reducing visual artifacts introduced by strided operations while maintaining the network structure.

**Benchmarking.** Zhou et al. (2016); Zhang et al. (2016) proposed a saliency-based evaluation where explanations are assessed based on how well they highlight complete instances of the classes of interest. Thus, treating model explanation as a weakly-supervised object detection/segmentation problem. This saliency-based protocol assumes that explanations are exclusive to intrinsic object features, e.g. color, shape, parts, etc. and completely ignores extrinsic features, e.g. scene, environment, related to the depicted context. Zeiler & Fergus (2014); Selvaraju et al. (2017) proposed a protocol based on crowd-sourced user studies. These type of evaluations are not only characterized by their high-cost, but also suffer from subjective bias (Gonzalez-Garcia et al. (2017)). Moreover, Das et al. (2016) suggest that deep models and humans do not necessarily attend to the same input evidence even when they predict the same output. Here we propose a protocol where the regions to

be highlighted by the explanation are predefined. The goal is to *objectify* the evaluation and relax the subjectivity introduced by human-based evaluations. Moreover, our protocol makes no strong assumption regarding the type of features highlighted by the the explanations.

## 3 PROPOSED METHOD

The proposed method consists of two parts. At training time, a set of relevant layer/filter pairs are identified for every class of interest $j$. This results in a relevance weight $w_j$, associated to class $j$, for every filter-wise response $x$ computed internally by the network. At test time, an image $I$ is pushed through the network producing the class prediction $\hat{j}=F(I)$. Then, taking into account the internal responses $x$, and relevance weights $w_{\hat{j}}$ for the predicted class $\hat{j}$, we generate visualizations indicating the image regions that contributed to this prediction.

### 3.1 IDENTIFYING RELEVANT FEATURES

One of the strengths of deep models is their ability to learn abstract concepts from simpler ones. That is, when an example is pushed into the model, a conclusion concerning a specific task can be reached as a function of the results (activations) of intermediate operations at different levels (layers) of the model. These intermediate results may hint at the "semantic" concepts that the model is taking into account when making a decision. From this observation, and the fact that activations are typically sparse, we make the assumption that some of the internal filters of a network encode features that are important for the task that the network addresses. To this end, we follow a procedure similar to Escorcia et al. (2015), aiming to predict each class $j$ by the linear combination $w_j \in \mathbb{R}^m$ of its internal activations $x$, with $m$ the total number of neurons/activations.

As an initial step, we extract the image-wise response $x_i$. To this end, we compute the $L_2$ norm of each channel (filter response) within each layer and produce a 1-dimensional descriptor by concatenating the responses from the different channels. This layer-specific descriptor is $L_1$-normalized in order to compensate for the difference in length among different layers. Finally, we concatenate all the layer-specific descriptors to obtain $x_i$. In this process, we do not consider the last layer whose output is directly related to the classes of interest, e.g. the last two layers from VGG-F Chatfield et al. (2014).

Following this procedure, we construct the matrix $X \in \mathbb{R}^{m \times N}$ by passing each of the $N$ training images through the network $F$ and storing the internal responses $x$. As such, the $i^{th}$ image of the dataset is represented by a vector $x_i \in \mathbb{R}^m$ defined by the filter-wise responses at different layers. Furthermore, the possible classes that the $i^{th}$ image belongs to are organized in a binary vector $l_i \in \{0,1\}^C$ where $C$ is the total number of classes. Putting the annotations from all the images together produces the binary label matrix $L=[l_1, l_2, ..., l_N]$, with $L \in \mathbb{R}^{C \times N}$. With these terms, we resort to solving the equation:

$$W^* = argmin_W \, ||X^T W - L^T||_F^2 \quad subject\ to: \, ||w_j||_1 \leq \mu \, , \, \forall_j = 1, ..., C \qquad (1)$$

with $\mu$ a parameter that allows controlling the sparsity. This is the matrix form of the $\mu$-lasso problem. This problem can be efficiently solved using the Spectral Gradient Projection method Mairal et al. (2014); van den Berg & Friedlander (2008). The $\mu$-lasso formulation is optimal for cases like the ones obtained by ResNet where the number of internal activations is large compared to the number of examples. After solving the $\mu$-lasso problem, we have a matrix $W=[w_1, w_2, ..., w_C]$, with $W \in \mathbb{R}^{m \times C}$. We impose sparsity on $W$ by enforcing the constraints on the $L_1$ norm of $w_j$, i.e., $||w_j||_1 \leq \mu \, , \, \forall_j = 1, ..., C$. As a result, each non-zero element in $W$ represents a pair of network layer $p$ and filter index $q$ (within the layer) of relevance.

### 3.2 GENERATING VISUAL FEEDBACK

During training time (Section 3.1), we identified a set of relevant features (indicated by $W$) for the classes of interest. At test time, we generate the feedback visualizations by taking into account the response of these features on the content of the tested images. Towards this goal, we push an image $I$ through the network producing the class prediction $\hat{j}=F(I)$. During that pass, we compute the internal filter-wise response vector $x_i$ following the procedure presented above. Then we compute the response $r_i^{\hat{j}}=(w_{\hat{j}} \circ x_i)$, where $\circ$ represents the element-wise product between two vectors. Note that the $w_{\hat{j}}$ vector is highly sparse, therefore adding an insignificant cost at test time. The features,

Figure 2: Heatmap visualization at lower layers of VGG-F . Note how our method attenuates the grid-like artifacts introduced by existing DeconvNet+GB methods (Springenberg et al. (2015)).

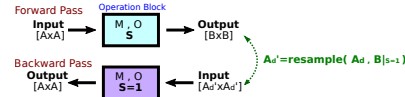

Figure 3: To attenuate artifacts , during the backward pass, we set the stride to 1 ($S = 1$) and compensate by resampling the input so that $A'_d = B|_{S=1}$.

i.e., layer/filter pairs $(p^*, q^*)$, with strongest contribution in the prediction $\hat{j}$ are selected as those with maximum response in $r_i^{\hat{j}}$. Finally, we feed this information to the Deconvnet-based method with guided backpropagation from Grün et al. (2016) to visualize the important features as defined by the layer/filter pairs $(p^*, q^*)$. Following the visualization method from Grün et al. (2016), given a filter $p$ from layer $q$ and an input image, we first push forward the input image through the network, storing the activations from each filter at each layer, until reaching the layer $p$. Then, we backpropagate the activations from filter $q$ at layer $p$ with inverse operations until reaching back to the input image space. As result we get as part of the output a set of heatmaps, associated to the relevant features, defined by $(p^*, q^*)$, indicating the influence of the pixels that contributed to the prediction. See Fig.1 (left) for an example of the visual feedback provided by our method. Please refer to Grün et al. (2016); Springenberg et al. (2015); Zeiler & Fergus (2014) for further details regarding Deconvnet-based and Guided backpropagation methods.

### 3.3 IMPROVING VISUAL FEEDBACK QUALITY

Deep neural networks addressing computer vision tasks commonly push the input visual data through a sequence of operations. A common trend of this sequential processing is that the input data is internally resampled until reaching the desired prediction space. As mentioned in Sec. 2, methods aiming at interpretation/explanation start from an internal point in the network and go backwards until reaching the input space - producing a heatmap. However, due to the resampling process, heatmaps generated by the backwards process tend to display grid-like artifacts. More precisely, we find that this grid effect is caused by the internal resampling introduced by network operations with stride larger than one ($S>1$). To alleviate this effect, in the backwards pass, we set the stride $S=1$ and compensate for this change by modifying the input accordingly. As a result, the backwards process can be executed while maintaining the network structure.

More formally, given a network operation block defined by a convolution mask with size $[M{\times}M]$, stride $[S, S]$, and padding $[O, O, O, O]$, the relationship between the size of its input $[A{\times}A]$ and its output $[B{\times}B]$ (see Fig. 3) is characterized by the following equation:

$$A + 2 \cdot O = M + (B - 1) \cdot S \tag{2}$$

from where,

$$B = [\,(A + 2 \cdot O - M)/S\,] + 1 \tag{3}$$

Our method starts from the input ($[A_d{\times}A_d]$), which encodes the contributions from the input image, carried by the higher layer in the Deconvnet backward pass. In order to enforce a "cleaner" resampling when $S>1$, during the backward pass, the size of the input ($[A_d{\times}A_d]$) of the operation block should be the same as that of the feature map ($[B{\times}B]$) produced by the forward pass if the stride $S$ were equal to one, i.e., $A'_d=B|_{S=1}$. According to Eq. 3 with $S=1$, $A_d$ should therefore be resampled to $A'_d=B|_{S=1}=A+2 \cdot O-M+1$. We do this resampling via the nearest-neighbor interpolation algorithm given its proven fast computation time which makes it optimal for real-time processing. By introducing this step, the network will perform the backwards pass at every layer with stride $S=1$ and the grid effect will disappear. See Fig. 2 for some examples of the improvements introduced by our method.

## 4 EVALUATION

We conduct four sets of experiments. First, in Sec. 4.1, we verify the importance of the identified relevant features in the task addressed by the network. Then, in Sec. 4.2, we evaluate the improvements on visual quality provided by our method. In Sec. 4.3, we quantify the capability of our visual explanations to highlight the regions that are descriptive for the classes of interest. Finally, in Sec. 4.4, we assess the sensitivity of the proposed method w.r.t. the predicted classes.

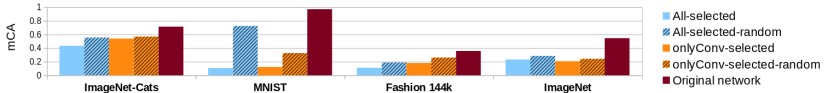

Figure 4: Changes in mean classification accuracy (mCA) as the identified relevant filters are ablated.

We evaluate the proposed method on an image recognition task. We conduct experiments on three standard image recognition datasets, i.e., MNIST LeCun & Cortes (2010), Fashion144k Simo-Serra et al. (2015) and imageNet (ILSVRC'12) Russakovsky et al. (2015). Additionally, we conduct experiments on a subset of cat images from imageNet (imageNet-cats). MNIST covers 10 classes of hand-written digits. It is composed by 70k images in total (60k for training/validation, 10k for testing). The imageNet dataset is composed of 1k classes. Following the standard practice, we measure performance on its validation set. Each class contains 50 validation images. For the Fashion144k dataset Simo-Serra et al. (2015), we consider the subset of 12k images from Wang et al. (2018) used for the geolocation of 12 city classes. The imageNet-cats subset consists of 13 cat classes, containing both domestic and wild cats. It is composed of 17,550 images. Each class contains 1,3k images for training and 50 images for testing. Please refer to the supplementary material for more implementation details.

### 4.1 IMPORTANCE OF IDENTIFIED RELEVANT FEATURES

In this experiment, we verify the importance of the "relevant" features identified by our method at training time (Sec. 3.1). To this end, given a set of identified features, we evaluate the influence they have in the network by measuring changes in classification performance caused by their removal. We remove features in the network by setting their corresponding layer/filter to zero. The expected behavior is that a set of features with high relevance will produce a stronger drop in performance when ablated. Fig. 4, shows the changes in classification performance for the tested datasets. We report the performance of four sets of features: a) *All*, selected with our method by considering the whole internal network architecture, b) *OnlyConv*, selected by considering only the convolutional layers of the network, c) a *Random* selection of features (filters) selected from the layers indicated in the sets a) and b), and for reference, d) the performance obtained by the original network. Note that the *OnlyConv* method makes the assumption that relevant features are only present in the convolutional layers. This is a similar assumption as the one made by state-of-the-art methods Bau et al. (2017); Xie et al. (2017); Zhou et al. (2016). When performing feature selection (Sec.3.1), we set the sparsity parameter $\mu=10$ for all the tested datasets. This produces subsets of 92|101, 46|28, 104|111, 248|180 relevant features for the *All*|*OnlyConv* methods, on the respective datasets from Fig. 4. Differences in the number of the selected features can be attributed to possibly redundant or missing predictive information between the initial pools of filter responses $x$ used to select the *All* and *OnlyConv* features.

A quick inspection of Fig. 4 shows that indeed classification performance drops when we remove the identified features, *All* and *OnlyConv*. Moreover, it is noticeable that a random removal of features has a lower effect on classification accuracy. This demonstrates the relevance of the identified features for the classes of interest. In addition, it is visible that the method that considers the complete internal structure, i.e., *All*, suffers a stronger drop in performance compared to the *OnlyConv* which only considers features produced by the convolutional layers. This suggests that there is indeed important information encoded in the fully connected layers, and while convolutional layers are a good source for features, focusing on them only does not reveal the full story. Regarding the effect of the sparsity value $\mu$ in the $\mu$-lasso formulation (Sec. 3.1), we note that increasing $\mu$ increases the number of selected features. This leads to more specialized features that can better cope with rare instances of the classes of interest. We decided to start from a relatively low value, e.g. $\mu=10$, in order to focus on a small set of relevant features that can generalize to the classes of interest while, at the same time, keeping the computational cost low.

**Qualitative Analysis.** In order to get a qualitative insight into the type of information that these features encode we compute an average visualization by considering the top 100 image patches where such features have a high response. Towards this goal, given the set of identified relevant features, for every class, we select images with higher responses. Then, we take the input image at the location with maximum response for a given filter and crop it by considering the receptive field of the corresponding layer/filter of interest. Selected examples of average images, with rich semantic representation, are presented in Fig. 5 for the tested datasets.

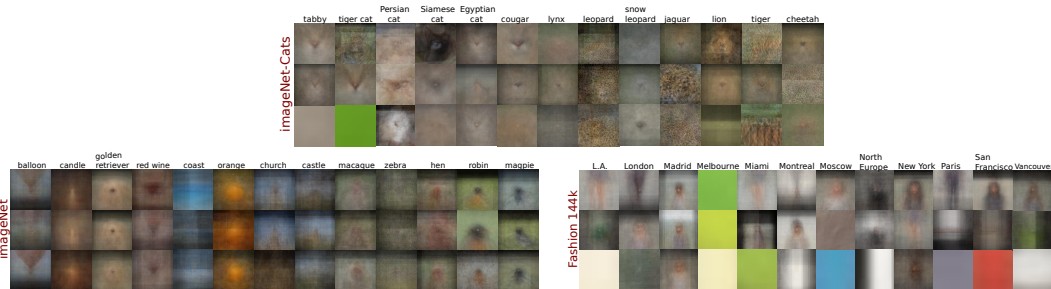

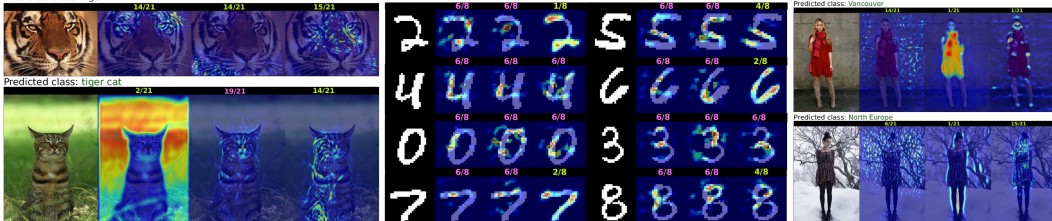

Figure 5: Average Images from the identified relevant filters for the ImageNet-Cats subset (top), some selected classes from the full ImageNet (left) and the Fashion144K (left) datasets, respectively.

Figure 6: Our visual explanations. We accompany the predicted class label with our heatmaps indicating the pixel locations, associated to the features, that contributed to the prediction. These features may come from the object itself as well as from its context. See how for MNIST, some features support the existence of gaps, as to avoid confusion with another class. On top of each heatmap we indicate the number of the layer where the features come from. The layer type is color-coded, i.e., convolutional (green) and fully connected (pink).

We can notice that for imageNet-Cats, the identified features cover descriptive characteristics of the considered cat classes. For example, the dark head of a Siamese cat, the nose/mouth of a cougar, or the fluffy-white body shape of Persian cat. Likewise, it effectively identifies the descriptive fur patterns from the jaguar, leopard and tiger classes and colors which are related to the background. We see a similar effect on a selection of other objects from the rest of the imageNet dataset. For instance, for scene-type classes, i.e., coast, castle and church, the identified features focus on the outline of such scenes. Similarly, we notice different viewpoints for animal-type classes, e.g. golden-retriever, hen, robin, magpie. On the Fashion144k dataset (Fig. 5 (right)) we can notice that some classes respond to features related to green, blue, red, and beige colors. Some focus on legs, covered and uncovered, while others focus on the upped body part. It is interesting that from the upper body parts, some focus on persons with dark long hair, short hair, and light hair. Similarly, there is a class with high response to horizontal black-white gradients where individuals tend to dress in dark clothes. These visualizations answer the question explored in Wang et al. (2018) and why the computer outperforms the surveyed participants. It shows that the model effectively exploits human-related features (legs clothing, hair length/color, clothing color) as well as background-related features, mainly covered by color/gradients and texture patterns. In the visual explanations provided by our method we can see that the model effectively uses this type of features to reach its decision.

Finally, in Fig. 6 we show some examples of the visual explanations produced by our method. We aggregate the predicted class label with our heatmap visualizations indicating the pixel locations, associated to the relevant features, that contributed to the prediction. For the case of the ILSVRC'12 and Fashion144k examples, we notice that the relevant features come from the object itself as well as from its context. For the case of the MNIST examples, in addition to the features firing on the object, there are features that support the existence of a gap (background), as to emphasize that the object is not filled there and avoid confusion with another class. For example, see for class 2 how it speaks against 0 and for 6 how it goes against 4.

## 4.2 VISUAL FEEDBACK QUALITY

In this section, we assess the visual quality of the visual explanations generated by our method. In Fig. 7, we compare our visualizations with upsampled activation maps from internal layers (Bau et al. (2017); Zhou et al. (2016)) and the output of DeconvNet with guided-backpropagation (Springenberg et al. (2015)). We show these visualizations for different layers/filters throughout the network.

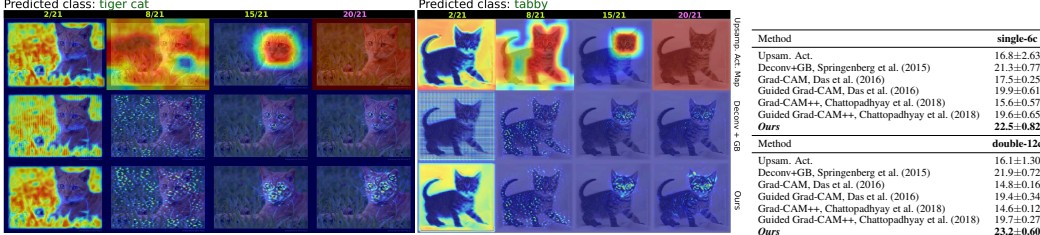

| Method | single-6c |
|---|---|
| Upsam. Act. | 16.8±2.63 |
| Deconv+GB, Springenberg et al. (2015) | 21.3±0.77 |
| Grad-CAM, Das et al. (2016) | 17.5±0.25 |
| Guided Grad-CAM, Das et al. (2016) | 19.9±0.61 |
| Grad-CAM++, Chattopadhyay et al. (2018) | 15.6±0.57 |
| Guided Grad-CAM++, Chattopadhyay et al. (2018) | 19.6±0.65 |
| *Ours* | **22.5±0.82** |

| Method | double-12c |
|---|---|
| Upsam. Act. | 16.1±1.30 |
| Deconv+GB, Springenberg et al. (2015) | 21.9±0.72 |
| Grad-CAM, Das et al. (2016) | 14.8±0.16 |
| Guided Grad-CAM, Das et al. (2016) | 19.4±0.34 |
| Grad-CAM++, Chattopadhyay et al. (2018) | 14.6±0.12 |
| Guided Grad-CAM++, Chattopadhyay et al. (2018) | 19.7±0.27 |
| *Ours* | **23.2±0.60** |

Figure 7: Pixel effect visualization for different methods. Note how for lower layers (8/21), our method attenuates the grid-like artifacts introduced by Deconvnet methods. For higher layers (15/21), our method provides a more precise visualization when compared to upsampled activation maps. For the case of FC layers (20/21), using upsampled activation maps is not applicable.

Table 1: Area under the IoU curve (in percentages) on an8Flower over *5-folds*.

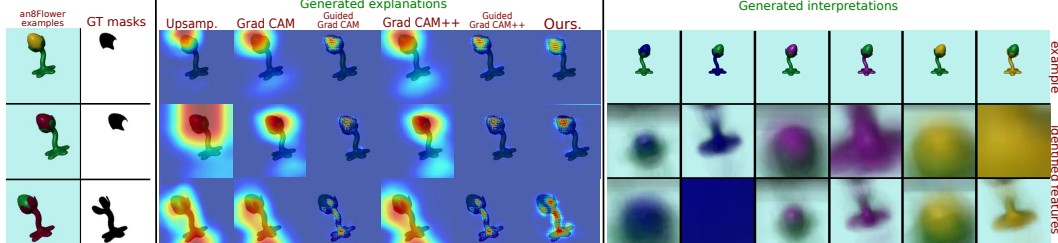

Figure 8: Left: Examples and GT-masks from the proposed *an8FLower* dataset. Center: Comparison of generated visual explanations. Right: Examples of the generated visual interpretations.

A quick inspection reveals that our method to attenuate the grid-like artifacts introduced by Deconvnet methods (see Sec 3.3) indeed produces noticeable improvements for lower layers. See Fig. 2 for additional examples presenting this difference at lower layers. Likewise, for the case of higher layers (Fig. 7), the proposed method provides more precise visualizations when compared to upsampled activation maps. In fact, the rough output produced by the activation maps at higher layers has a saliency-like behavior that gives the impression that the network is focusing on a larger region of the image. This could be a possible attribution to why in earlier works Zhou et al. (2015), manual inspection of network activations suggested that the network was focusing on "semantic" parts. Please see Gonzalez-Garcia et al. (2017) for an in-depth discussion of this observation. Finally, for the case of FC layers, using upsampled activation maps is not applicable. Please refer to the supplementary material for additional examples. In addition, to quantitatively measure the quality of our heatmaps we perform a box-occlusion study Zeiler & Fergus (2014). Given a specific heatmap, we occlude the original image with patches sampled from the distribution defined by the heatmap. We measure changes in performance as we gradually increase the number of patches up to covering the 30% most relevant part of the image. Here our method reaches a mean difference in prediction confidence of 2% w.r.t. to Springenberg et al. (2015). This suggests that our method is able to maintain focus on relevant class features while producing detailed heatmaps with better visual quality.

## 4.3 MEASURING VISUAL EXPLANATION ACCURACY

We generate two synthetic datasets, *an8Flower-single-6c* and *an8Flower-double-12c*, with 6 and 12 classes respectively. In the former, a fixed single part of the object is allowed to change color. This color defines the classes of interest. In the latter, a combination of color and the part on which it is located defines the discriminative feature. After defining these features, we generate masks that overlap with the discriminative regions (Fig. 8 (left)). Then, we threshold the heatmaps at given values and measure the pixel-level intersection over union (IoU) of a model explanation (produced by the method to be evaluated) w.r.t. these masks. We test a similar model as for the MNIST dataset (Sec. 4.1) trained on each variant of the *an8Flower* dataset. In Table 1 we report 5-fold cross-validation performance of the proposed feature selection method using three different means (Upsamp. Act. Maps, Deconv+GB Springenberg et al. (2015) and ours heatmap variant) and other state-of-the-art methods to generate visual explanations.

We can notice in Fig. 8 (right) that our method effectively identifies the pre-defined discriminative regions regardless of whether they are related to color and/or shape. Likewise, Fig. 8 (center) shows that our explanations accurately highlight these features and that they have a better balance between

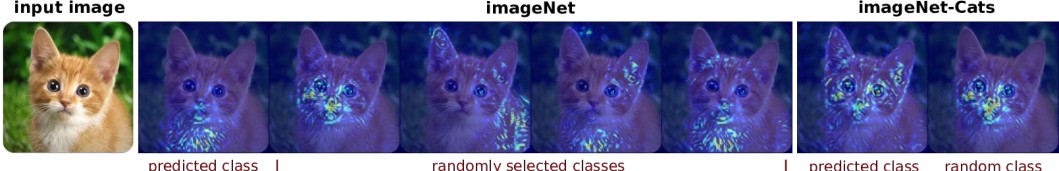

Figure 9: Sensitivity of the proposed method w.r.t. the predicted class. Note how the generated explanation focuses on different regions of the input image (left) when explaining different classes. We see a similar trend on explanations generated from the models trained on the imageNet (center) and imageNet-cats (right) datasets.

level of detail and coverage than those produced by existing methods. The quantitative results (Table 1) show that our method has a higher mean IoU of the discriminative features when compared to existing methods. However, it should be noted that, different from the compared methods, our method involves an additional process, i.e., feature selection via $\mu$-lasso, at training time. Moreover, for this process an additional parameter, i.e $\mu$, should be defined (Sec. 3.1). Please refer to the supplementary material for more details and visualizations.

Methods for model explanation/interpretation aim at providing users with insights on what a model has learned and why it makes specific predictions. Putting this together with the observations made in our experiments, there are two points that should be noted. On the one hand, we believe that our objective evaluation should be complemented with simpler user studies. This should ensure that the produced explanations are meaningful to the individuals they aim to serve. On the other hand, our proposed evaluation protocol enables objective quantitative comparison of different methods for visual explanation. As such it is free of the weaknesses of exhaustive user studies and of the complexities that can arise when replicating them.

### 4.4 Checking the Sanity of the Generated Visual Explanations

Beyond the capability of generating accurate visual explanations, recent works (Kindermans et al. (2017); Julius Adebayo (2018); Nie et al. (2018)) have stressed the importance of verifying that the generated explanations are indeed relevant to the model and the classes being explained. Towards this goal, we run a similar experiment to that conducted in Nie et al. (2018) where the visual explanation produced for a *predicted class* of a given model after observing a given image is compared against those when a *different class* is considered when generating the explanation. A good explanation should be sensible to the class, and thus generate different visualizations. In Fig. 9 we show qualitative results obtained by running this experiment on our models trained on the ILSVRC'12 (Russakovsky et al. (2015)) and imageNet-cats datasets.

As can be noted in Fig. 9, the explanation generated for the predicted class, i.e., 'cat'/'tabby', focuses on different regions than those generated for randomly selected classes. This is more remarkable for the case of the imageNet-cats model, which can be considered a fine-grained classification task. In this setting, when changing towards a random class, i.e., 'jaguar, panther, Panthera onca, Felis onca', the generated explanations only highlight the features that are common between the random class and the 'tabby' class depicted in the image. In their work, Nie et al. (2018) and Julius Adebayo (2018) found that explanations from DeconvNet and Guided-Backpropagation methods are not performing well in this respect, yielding visualizations that are not determined by the predicted class, but by the filters of the first layer and the edge-like structures in the input images. Although our method relies on DeconvNet and Guided-Backpropagation, our explanations go beyond regions with prominent gradients - see Fig. 1, 6 & 8. In fact, in classes where color is a discriminative feature, uniform regions are highlighted. This different result can be understood since, in our method, DeconvNet with Guided-Backpropagation is merely used as a means to highlight the image regions that justify the identified relevant features, not the predicted classes themselves. If a better, more robust or principled visualization method is proposed in the future by the community, we could use that as well.

## 5 Conclusion

We propose a method to enrich the prediction made by DNNs by indicating the visual features that contributed to such prediction. Our method identifies features encoded by the network that are relevant for the task addressed by the DNN. It allows *interpretation* of these features by the generation of average feature-wise visualizations. In addition, we proposed a method to attenuate

the artifacts introduced by strided operations in visualizations made by Deconvnet-based methods. This empowers our method with richer visual feedback with pixel-level precision without requiring additional annotations for supervision. Finally, we have proposed a novel dataset designed for the objective evaluation of methods for explanation of DNNs.

ACKNOWLEDGMENTS

This work was supported by the FWO SBO project Omnidrone, the VLAIO RD-project SPOTT , the KU Leuven PDM Grant PDM/16/131, and a NVIDIA GPU grant.

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

# 6 SUPPLEMENTARY MATERIAL

This section constitutes supplementary material. As such, this document is organized in six parts. In Section 6.1, we provide implementation details of the proposed method and experiments presented in the original manuscript. In Section 6.2, we provide a further quantitative analysis on the importance of the identified relevant features. In Section 6.3, we provide additional details regarding the generation of *an8Flower*, a dataset specially designed for evaluating methods for visual explanation. In Section 6.4, we provide extended examples on the average images used for interpretation. Similarly, in Section 6.5, we provide additional examples of visual explanations provided by our method. Finally, we concluded this document in Section 6.6 by performing a qualitative comparison in order to display the advantages of the proposed method over existing work.

## 6.1 IMPLEMENTATION DETAILS

We use in our experiments the pre-trained models provided as part of the MatconvNet framework Vedaldi & Lenc (2015) for both the MNISTand ImageNet datasets. For the Fashion144k dataset we use the VGG-F -based model (*Finetunned with image-based pooling*) released by the authors from Wang et al. (2018). For MNIST, we employ a network composed by 8 layers in total, five of them are convolutional , two are fully connected. The last one is a softmax layer. For the full imageNetset, we employ a VGG-F Chatfield et al. (2014) model which is composed of 21 layers, from these 15 are convolutional followed by five fully connected. The last one is a softmax layer. Finally, for the case of the imageNet-Cats subset we finetune the VGG-F model trained on the full imageNetset.

## 6.2 IMPORTANCE OF THE IDENTIFIED FEATURES

In order to verify the relevance of the identified features, i.e., how well the features encode information from the classes of interest, we measure the level to which they are able to "reconstruct" each of the classes of interest. Towards this goal, we compute the mean area under the ROC curve (mean-AUC) for all the classes of interest in a given dataset. In Figure 10, we report performance over different $\mu$ values. It can be noted that already with a low amount of selected features, i.e.. a low value of $\mu$, we can already encode, properly, visual characteristics of the classes of interest. This shows that the identified features hold strong potential as visual means for explanation.

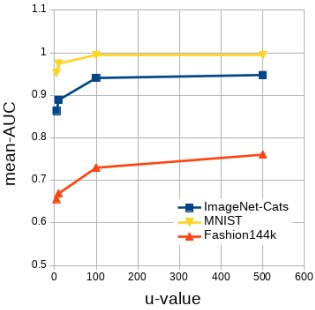

Figure 10: Classification based on the identified relevant features. We present the mean area under the ROC curve (mean-AUC) for all the classes of interest in a given dataset. Note that only a small amount of features, i.e., low $\mu$, is required to produce a good reconstruction.

## 6.3 AN8FLOWER DATASET

We release an8Flower, a dataset and evaluation protocol specifically designed for evaluating the performance of methods for model explanation. We generate this dataset by taking as starting point the eggplant model [2] publicly released with the Anim8or [3] 3D modeling software. Then, we introduce into this model the discriminative features to define each of the classes of interest. For each class of interest, we rotate the object 360 degrees and render/save 40 frames at different viewpoints with a size of 300x300 pixels.

Afterwards, in order to increase variation and the amount of data, we apply the following data augmentation procedure on each image frame. For each of the original 300x300 rendered frames, we crop each image five times at four corners and center respectively with the size of 250x250. Then, each cropped image is rotated five angles: 5, 10, 15, 20 and 25 degrees. In the end, the data augmentation produces a total of 1000 example images

---

[2]http://www.anim8or.com/learn/tutorials/eggplant/index.html
[3]http://www.anim8or.com/

an8flower-single-6c

an8flower-double-12c

an8flower-part-2c

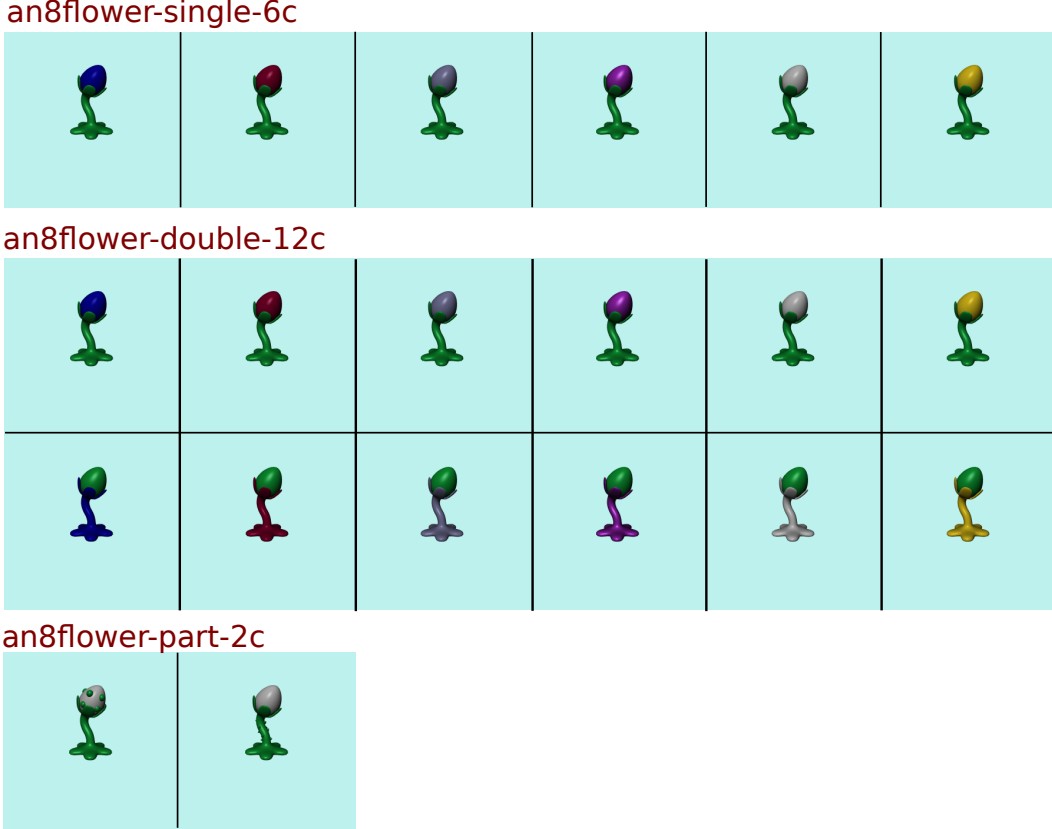

Figure 11: Rendered images from three variant of the proposed **an8Flower** dataset. *an8Flower-single-6c* contains six different flower colors, *an8Flower-double-12c* has six more classes, the same six colors but focused on the stem part. *an8Flower-part-2c* consists of two classes defined by the occurrence of balls on the flower or thorns on the stem part.

per class. Figure 11 shows rendered images of the classes of interest considered in three variants of this dataset, i.e., *an8Flower-single-6c*, *an8Flower-double-12c* and *an8Flower-part-2c*. Figure 12 and Figure 13 displays some examples of the augmented data used in training/testing and its corresponding mask.

## 6.4 VISUAL INTERPRETATION

In this section, we provide extended examples of the average images used by our method as means for visual interpretation *(Section 4.1 of the original manuscript)*. Moreover, we provide a visual comparison with their counterparts generated by existing methods. More precisely, up-scaled activations maps from convolutional layers Bau et al. (2017); Zhou et al. (2015) and heatmaps generated from deconvnets with guided backpropagation Grün et al. (2016); Springenberg et al. (2015); Zeiler & Fergus (2014).

In Figure 14, we show average images for the imageNet-Cats Russakovsky et al. (2015) subset, and the Fashion144k Simo-Serra et al. (2015) datasets, respectively. For each class on each dataset, we show the average image of each identified feature sorted, from top to bottom, based on their relevance for its corresponding class. In a similar fashion, in Figure 15, we show a visualization for the displayed subset of classes *(Figure 5 (center) in the original manuscript)* from the imageNet dataset. Given that for the imageNet dataset most of the identified features come from fully connected (FC) layers, no up-scaled response map visualization is possible for them. Therefore, we opted not to display average visualizations based on up-scaled activation maps. Finally, in Figure 16, we show average visualizations for additional classes from imageNet.

In Figure 14 we can notice that already within few top relevant features per class, semantic concepts start to appear. In addition, we see the same trends observed in the original paper. On the one hand, some features encode class-specific properties, e.g. nose shape, fur pattern, for the case of the imageNet-Cats, or hair color/length, leg dressing, or clothing color for the case of Fashion 144k. On the other hand, some features encode properties related to the background/context in which the images are captured, e.g. wall/road colors, vegetation, etc. We

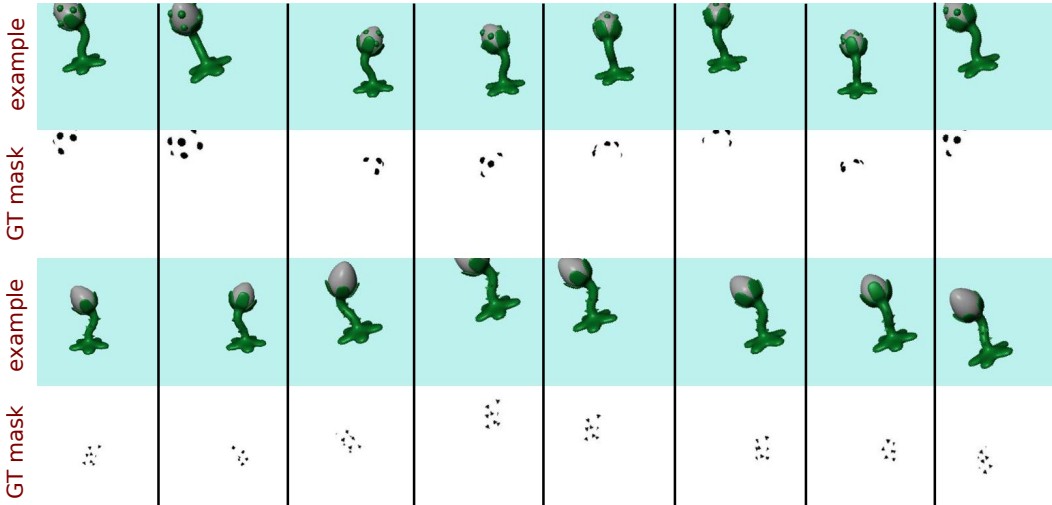

Figure 12: Random selection of examples from the *an8Flower-part-2c* with their corresponding ground-truth masks.

can notice that for the case of up-sampled activation maps, there are identified features which originate in Fully Connected Layers (FC). For this type of layer, the up-scaling process is not applicable.

When comparing the average visualizations generated by the different methods, it is noticeable that the proposed method produces sharper visualizations than those generated from up-scaled activation maps. Moreover, our visualizations are still sharper than those produced by state-of-the-art deconvnet-based methods with guided-backpropagation Springenberg et al. (2015). Therefore, enhancing the interpretation capabilities of the proposed method.

## 6.5 VISUAL EXPLANATION

In Figures 17, 18, 19, 13, 12 we extend the visual explanation results presented in the original manuscript. *(Figures 1, 6, 7 and 9 in the original manuscript)*

In the visual explanations generated by our method we accompany the predicted class label with our heatmaps indicating the pixel locations, associated to the features, that contributed to the prediction. In line with the original manuscript, on top of each heatmap we indicate the number of the layer where the features come from.

## 6.6 VISUAL QUALITY COMPARISON

We conclude this document by providing extended results related to the visual quality comparison *(Section 4.2 in the original manuscript)* of the visualizations generated by our method. Towards this goal, in Figures 23-27 we compare our visualizations with upsampled activation maps from internal layers Bau et al. (2017); Zhou et al. (2015) and the output of deconvnet combined with guided-backpropagation Springenberg et al. (2015). Following the same methodology as in Figure 2 of the original manuscript, we focus on visualizations at lower and higher layers of the network, i.e., layer-2/21 and layer-15/21, respectively.

For reference, in Figure 28 we show similar comparisons from explanations generated from models trained on the proposed *an8Flower* dataset.

In Figures 23-27 we can corroborate that our method to attenuate the grid-like artifacts introduced by deconvnet methods indeed produces noticeable improvements, for lower layers. Likewise, for the case of higher layers, the proposed method provides a more precise visualization when compared to upsampled activation maps.

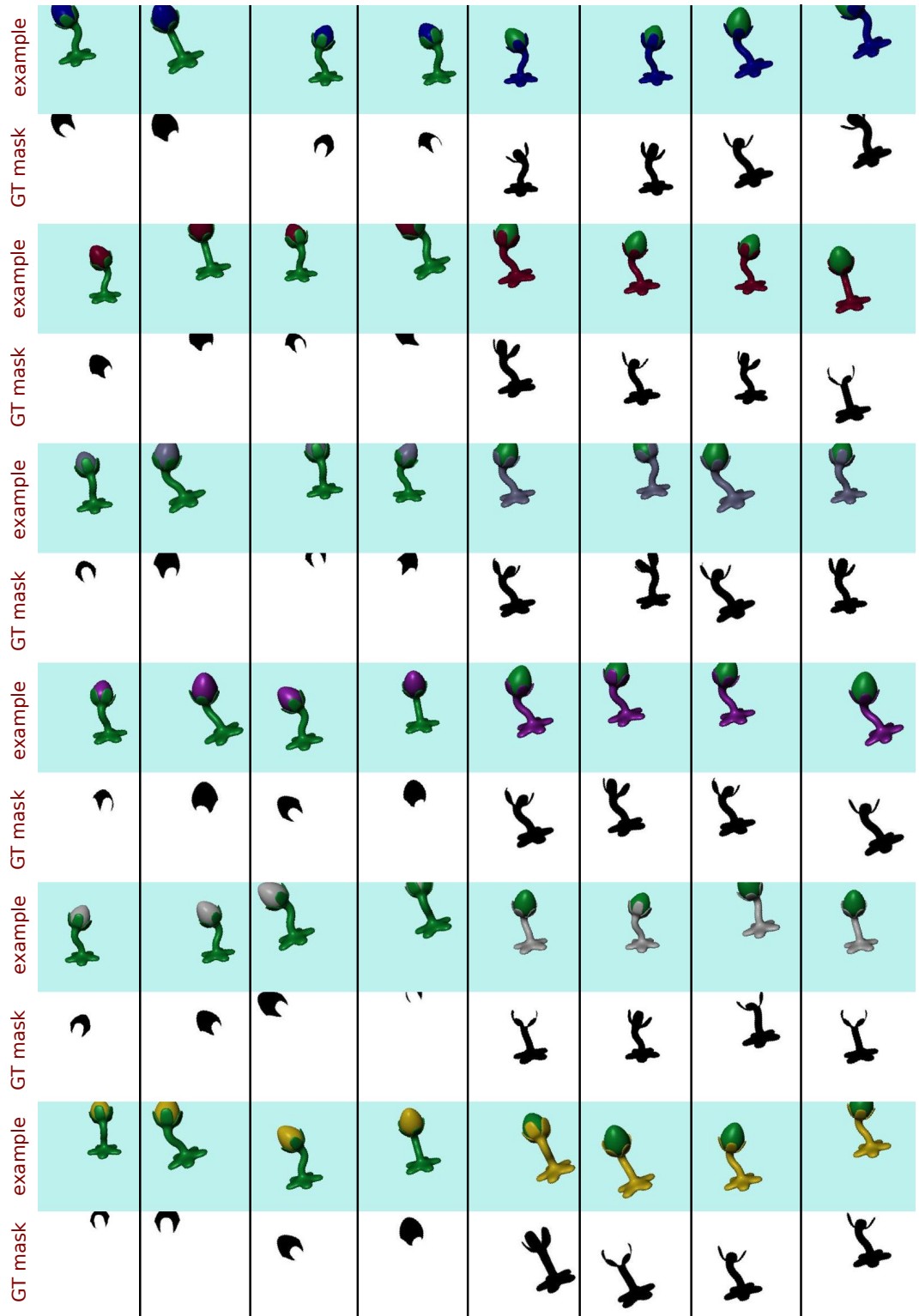

Figure 13: Random selection of examples from the *an8Flower-single-6c* and *an8Flower-double-12c* with their corresponding ground-truth masks

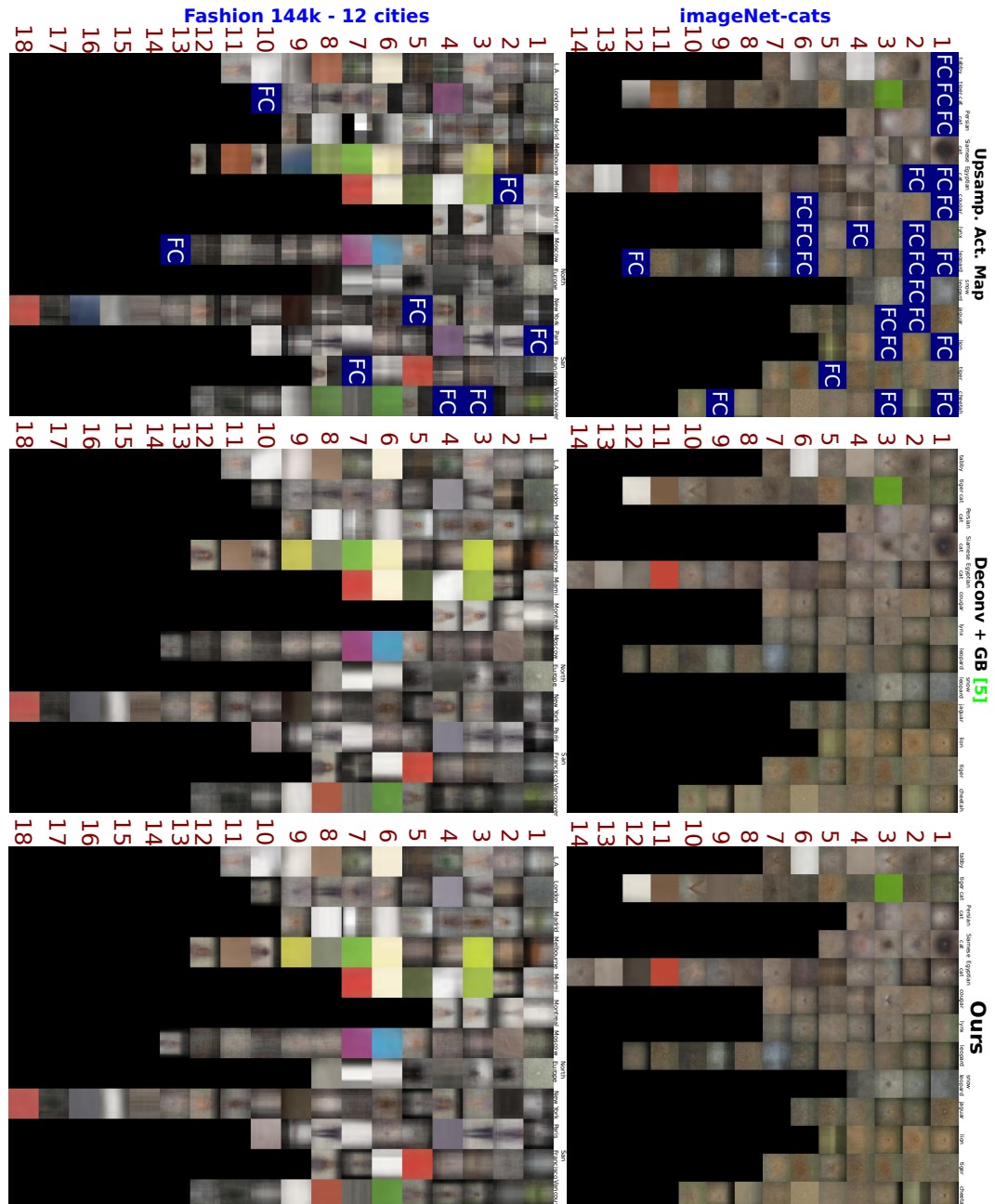

Figure 14: **Interpretation** (Average) visualizations for the identified relevant features for the **imageNet-Cats** Russakovsky et al. (2015) and a subset of the **Fashion144k** Simo-Serra et al. (2015) dataset. For each class on each dataset, average features are sorted by decreasing relevance in the class they encode. Average images are generated by either considering: up-scaled activation maps, heatmaps from methods based on deconvnet with guided-backpropagation (deconv+GB) Springenberg et al. (2015), or our method.

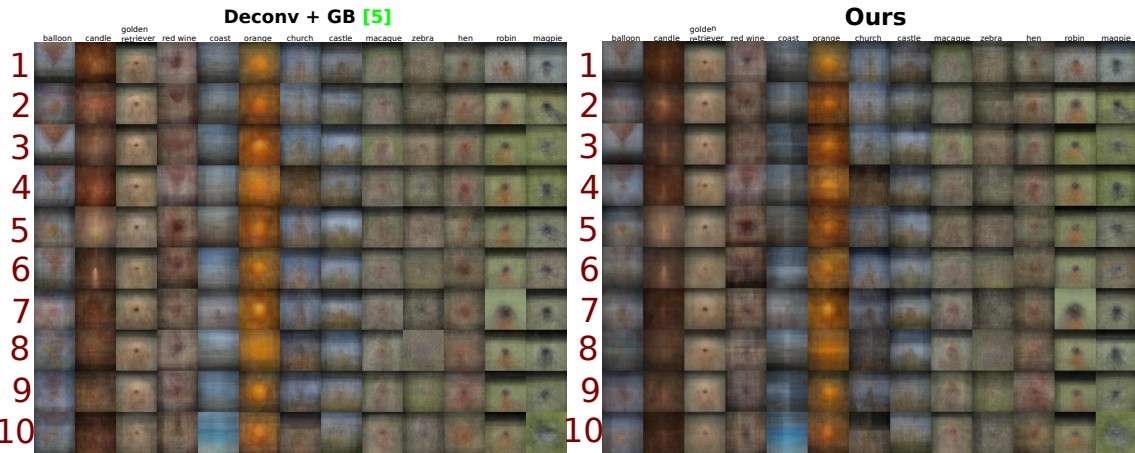

Figure 15: **Interpretation** (Average) visualizations for the identified relevant features for a subset of classes from the **imageNet** Russakovsky et al. (2015) dataset. Average images are generated by either considering heatmaps from methods based on deconvnet with guided-backpropagation (deconv+GB) Springenberg et al. (2015) (left) and our method (right).

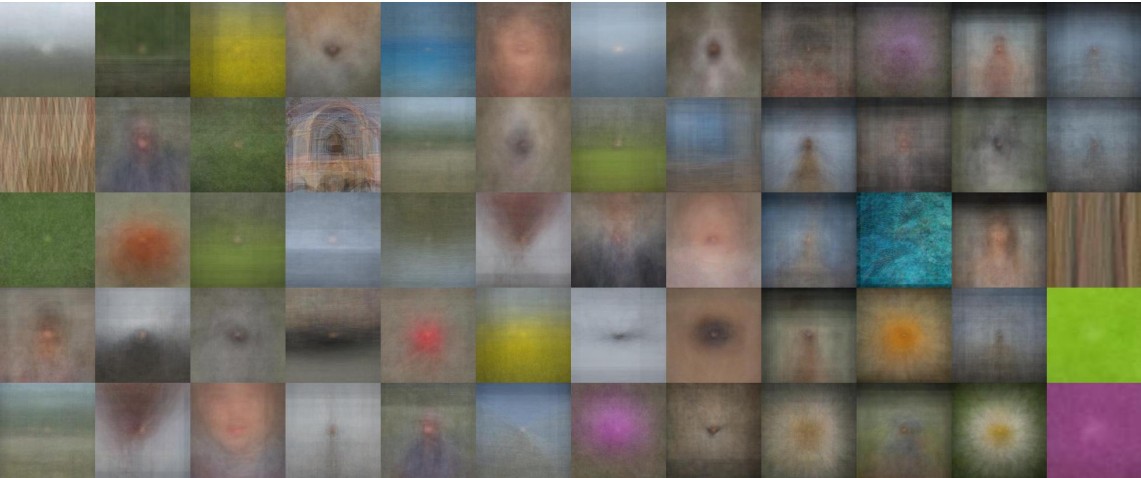

Figure 16: **Interpretation** (Average) visualizations produced by our method for the identified relevant features for the **imageNet** Russakovsky et al. (2015) dataset.

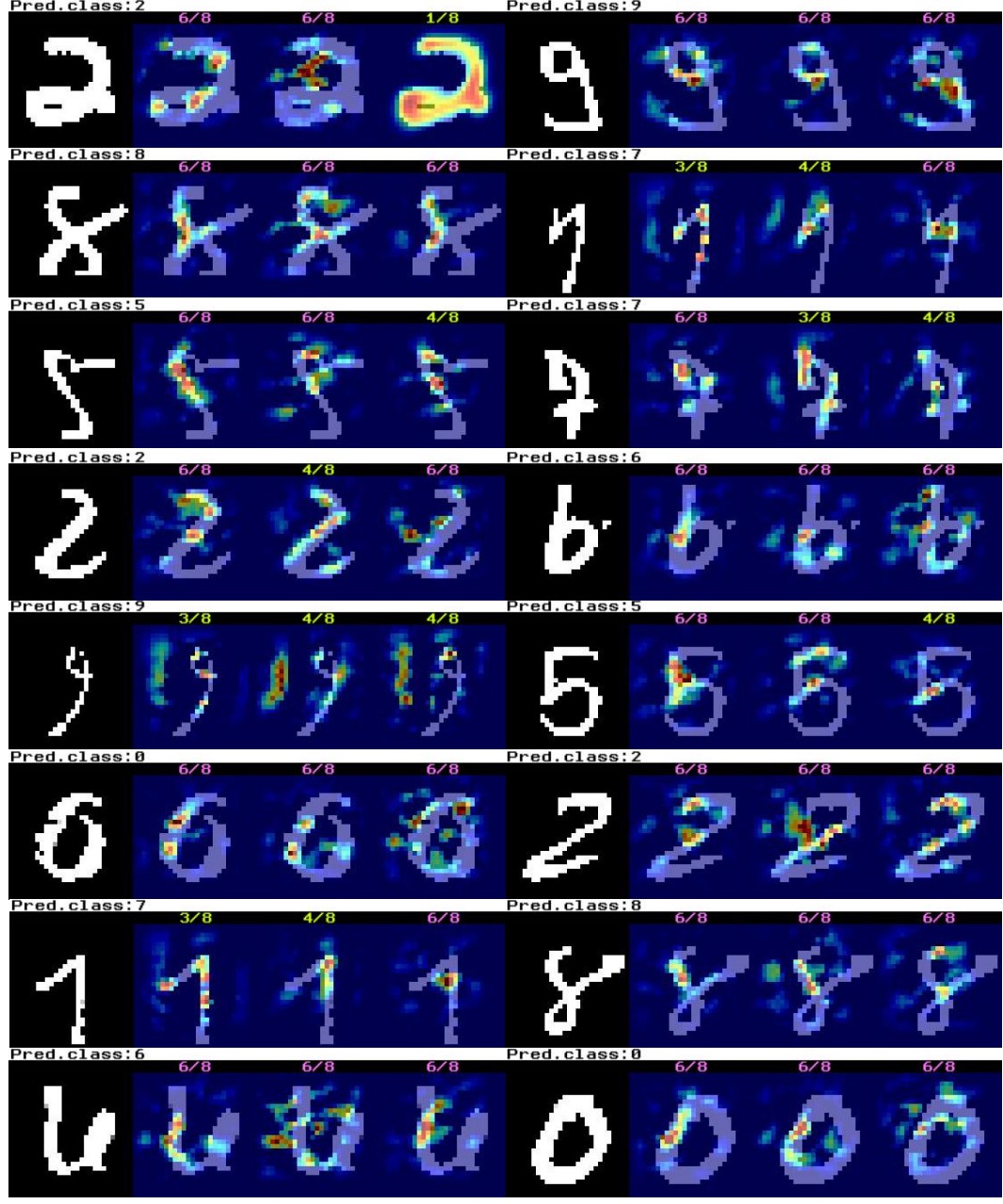

Figure 17: **Generated visual explanations** from the **MNIST** dataset. We accompany the predicted class label with our heatmaps indicating the pixel locations, associated to the features, that contributed to the prediction. On top of each heatmap we indicate the number of the layer where the features come from. The layer type is color-coded, i.e., convolutional (green) and fully connected (pink).

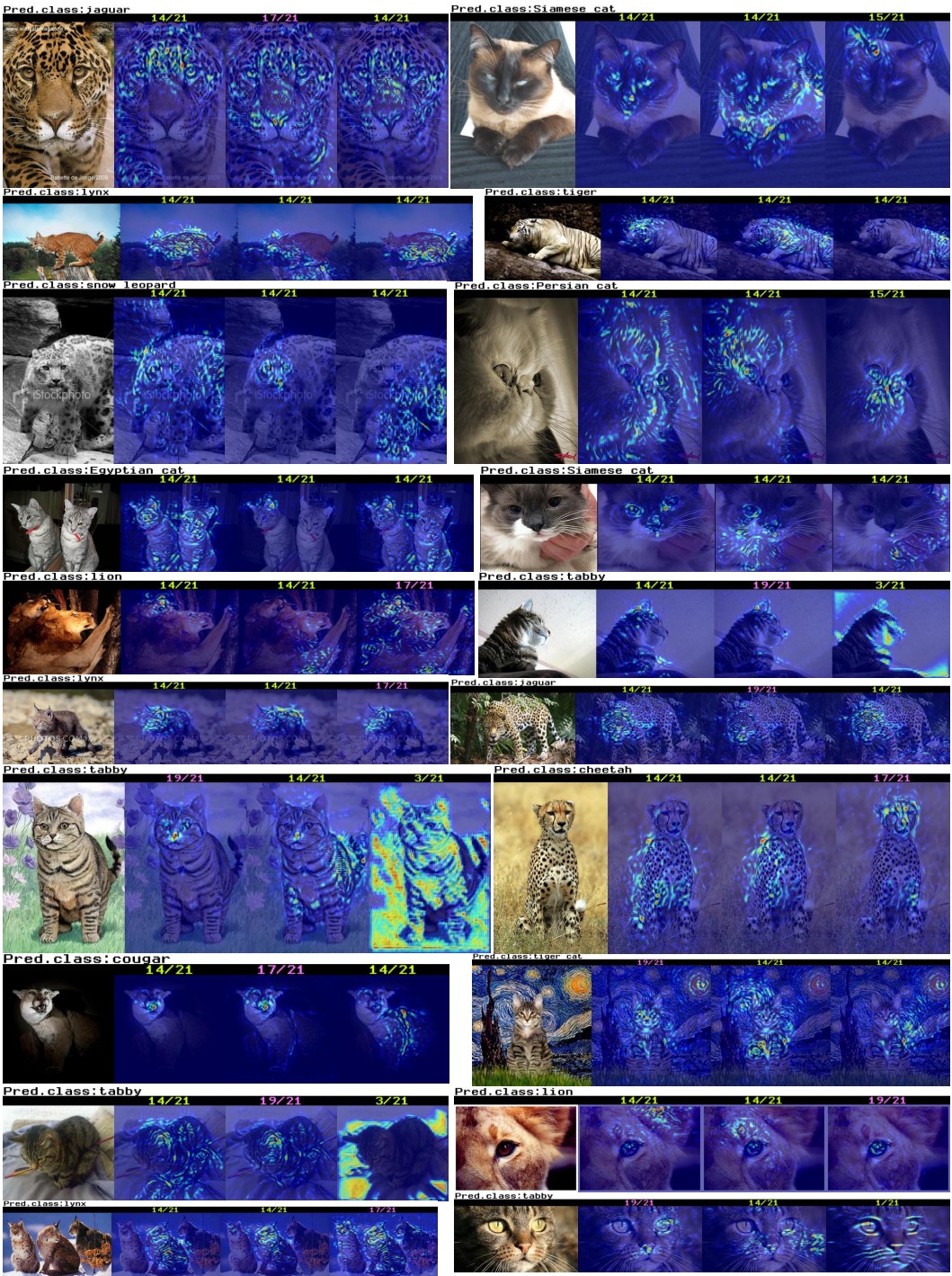

Figure 18: **Generated visual explanations** from the **imageNet-Cats** Russakovsky et al. (2015) subset. We accompany the predicted class label with our heatmaps indicating the pixel locations, associated to the features, that contributed to the prediction. On top of each heatmap we indicate the number of the layer where the features come from. The layer type is color-coded, i.e., convolutional (green) and fully connected (pink).

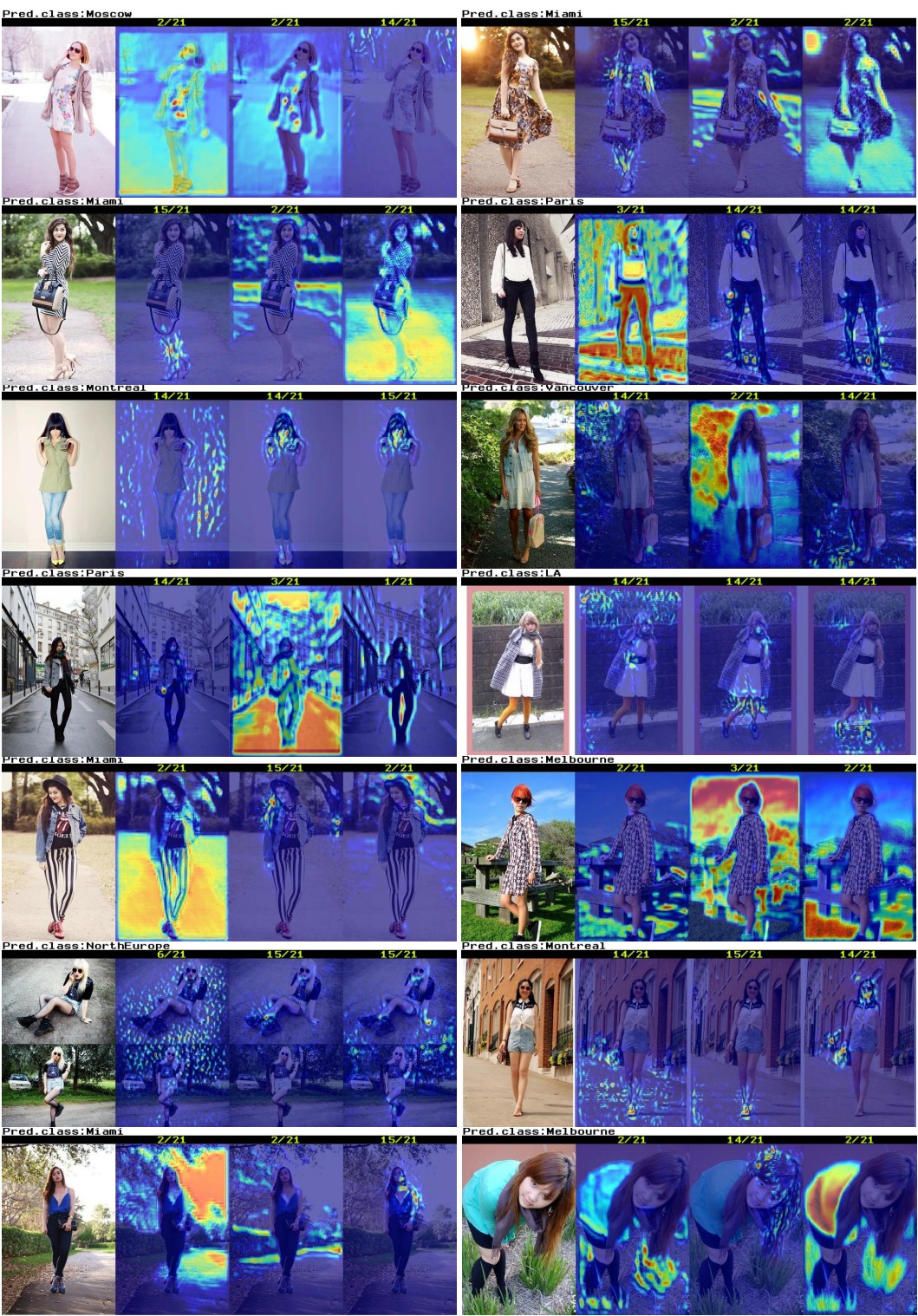

Figure 19: **Generated visual explanations** from the **Fashion144k** dataset Simo-Serra et al. (2015). We accompany the predicted class label with our heatmaps indicating the pixel locations, associated to the features, that contributed to the prediction. On top of each heatmap we indicate the number of the layer where the features come from. The layer type is color-coded, i.e., convolutional (green) and fully connected (pink).

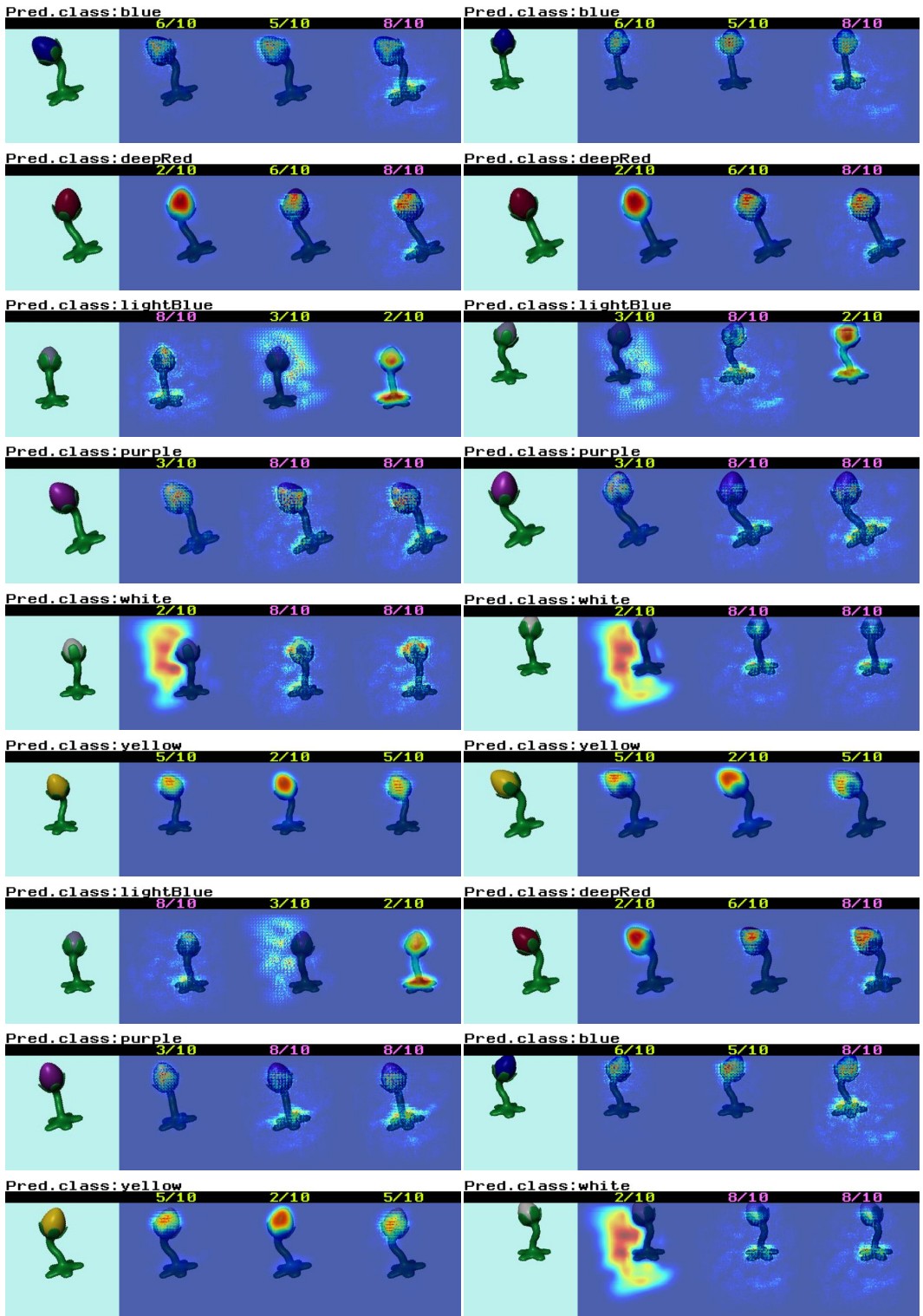

Figure 20: **Generated visual explanations** from our proposed **An8Flower-single-6c** dataset. We accompany the predicted class label with our heatmaps indicating the pixel locations, associated to the features, that contributed to the prediction. On top of each heatmap we indicate the number of the layer where the features come from. The layer type is color-coded, i.e., convolutional (green) and fully connected (pink).

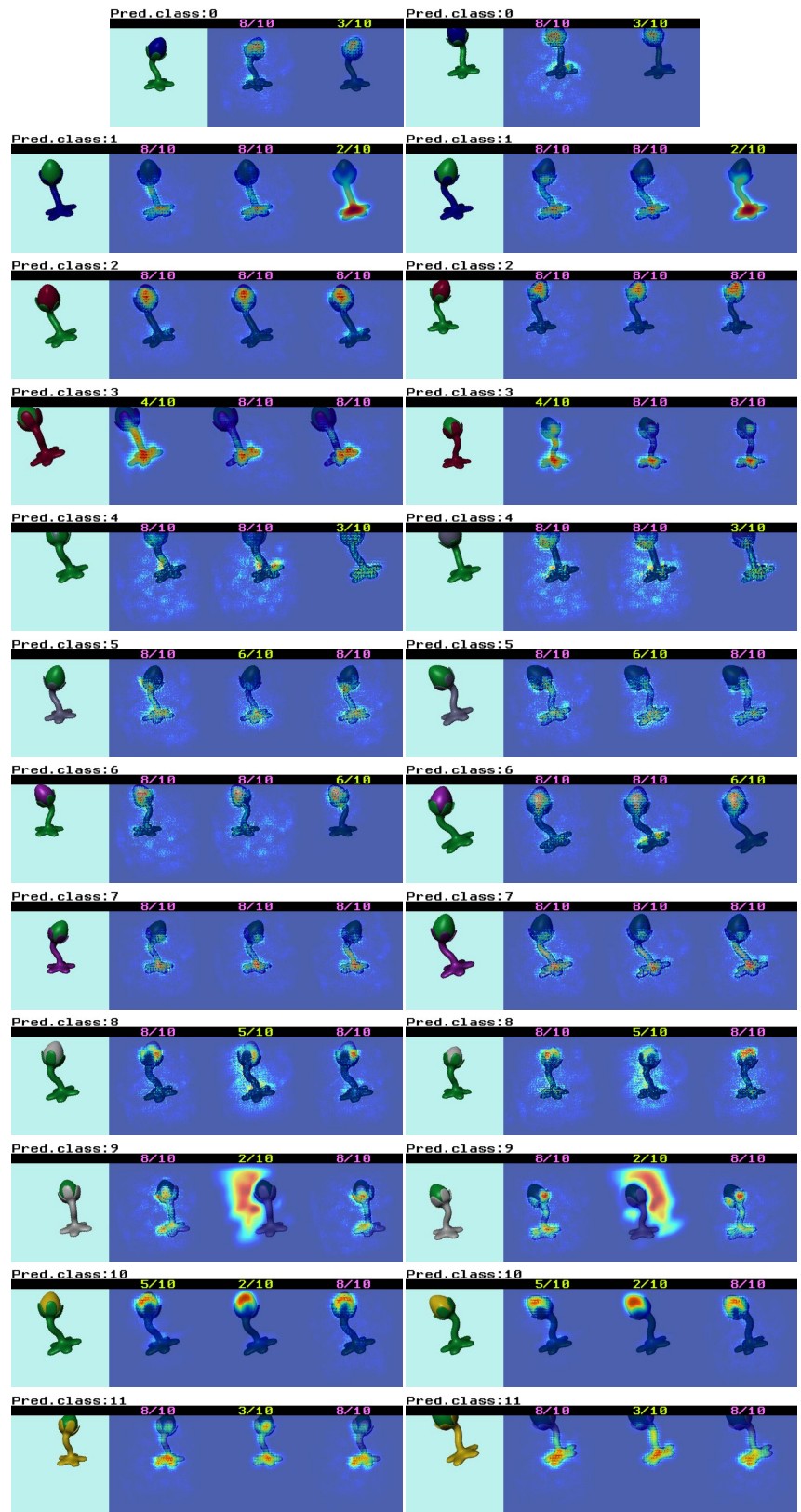

Figure 21: **Generated visual explanations** from our proposed **An8Flower-double-12c** dataset. We accompany the predicted class label with our heatmaps indicating the pixel locations, associated to the features, that contributed to the prediction. On top of each heatmap we indicate the number of the layer where the features come from. The layer type is color-coded, i.e., convolutional (green) and fully connected (pink).

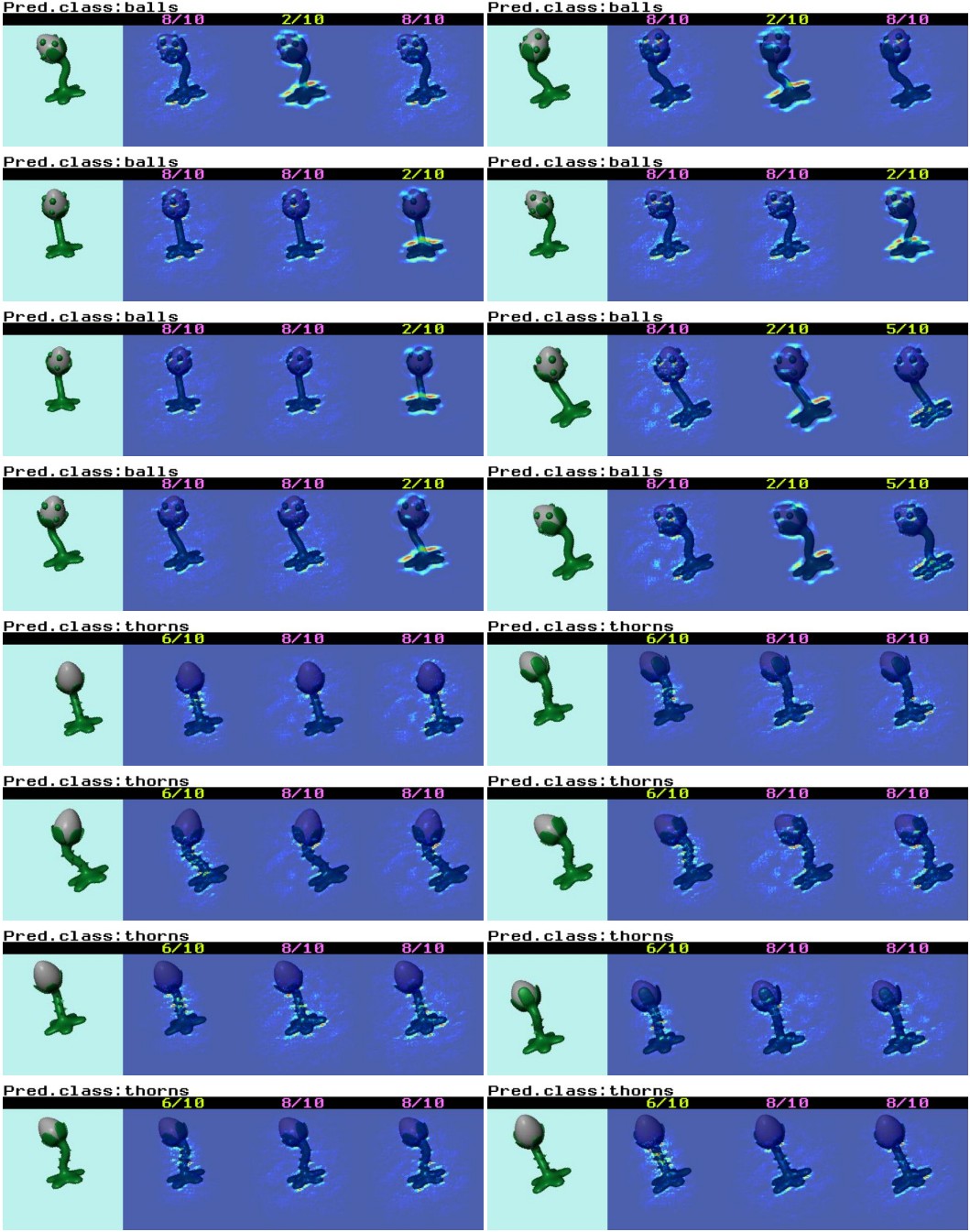

Figure 22: **Generated visual explanations** from our proposed **An8Flower-part-2c** dataset. We accompany the predicted class label with our heatmaps indicating the pixel locations, associated to the features, that contributed to the prediction. On top of each heatmap we indicate the number of the layer where the features come from. The layer type is color-coded, i.e., convolutional (green) and fully connected (pink).

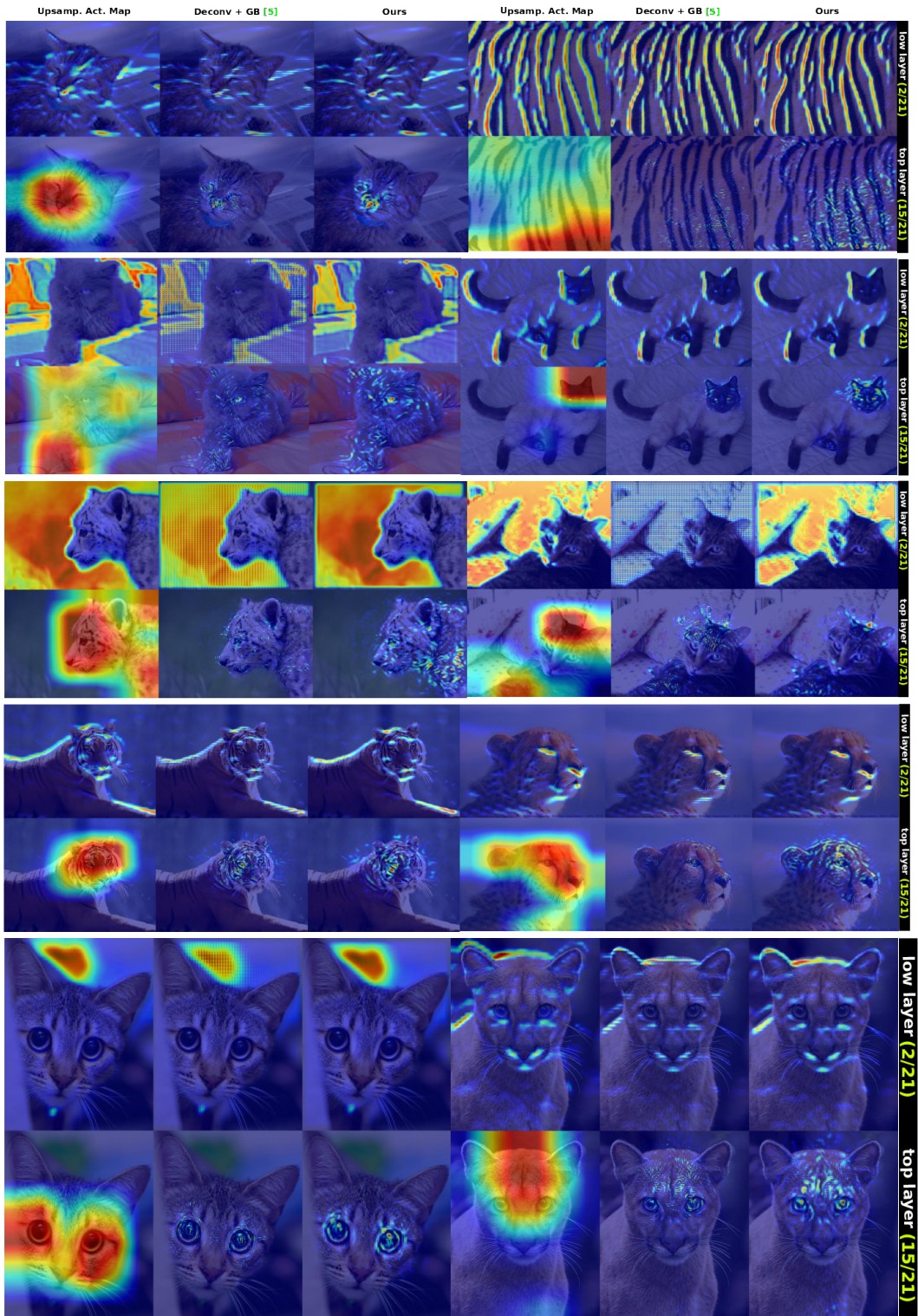

Figure 23: **Visual quality comparison** for visualizations generated from the **imageNet-Cats** subset Russakovsky et al. (2015). Note how our heatmaps attenuate the grid-like artifacts introduced by deconvnet-based methods at lower layers. Likewise, our method is able to produce a more detailed visual feedback than upsampled activation maps.

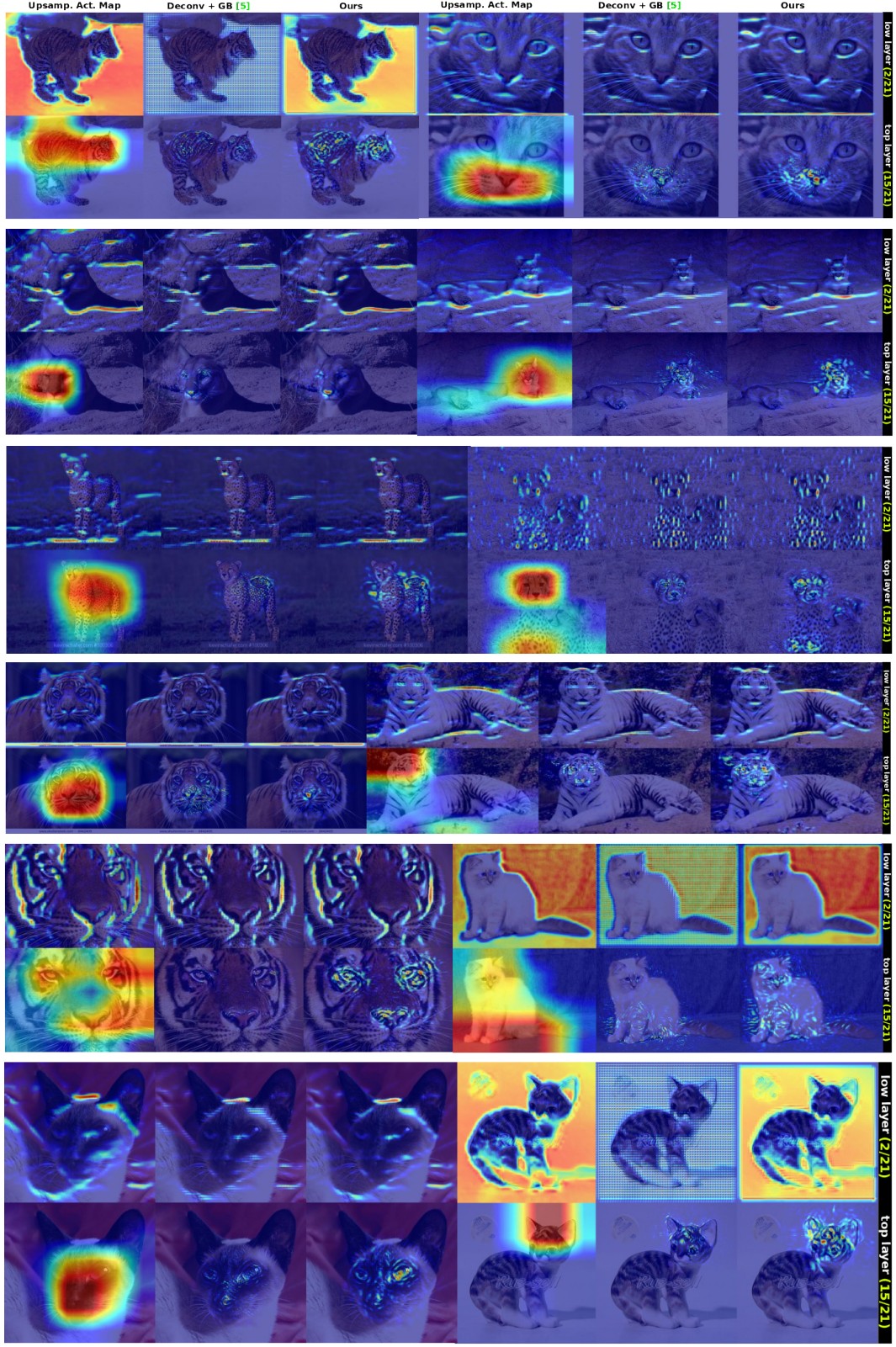

Figure 24: **Visual quality comparison** for visualizations generated from the **imageNet-Cats** subset Russakovsky et al. (2015). Note how our heatmaps attenuate the grid-like artifacts introduced by deconvnet-based methods at lower layers. Likewise, our method is able to produce a more detailed visual feedback than upsampled activation maps.

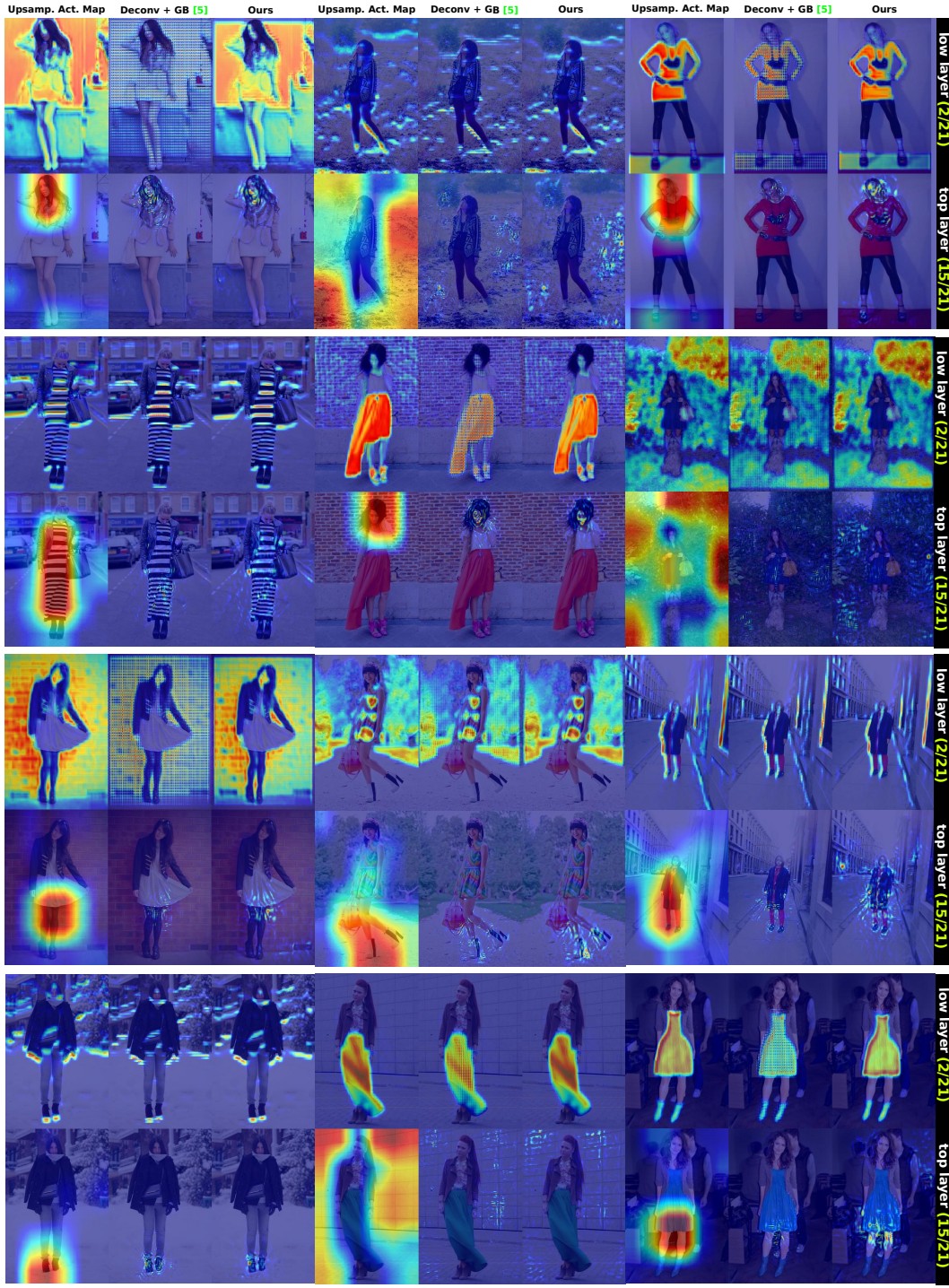

Figure 25: **Visual quality comparison** for visualizations generated from the **Fashion114k** dataset Simo-Serra et al. (2015). Note how our heatmaps attenuate the grid-like artifacts introduced by deconvnet-based methods at lower layers. Likewise, our method is able to produce a more detailed visual feedback than upsampled activation maps.

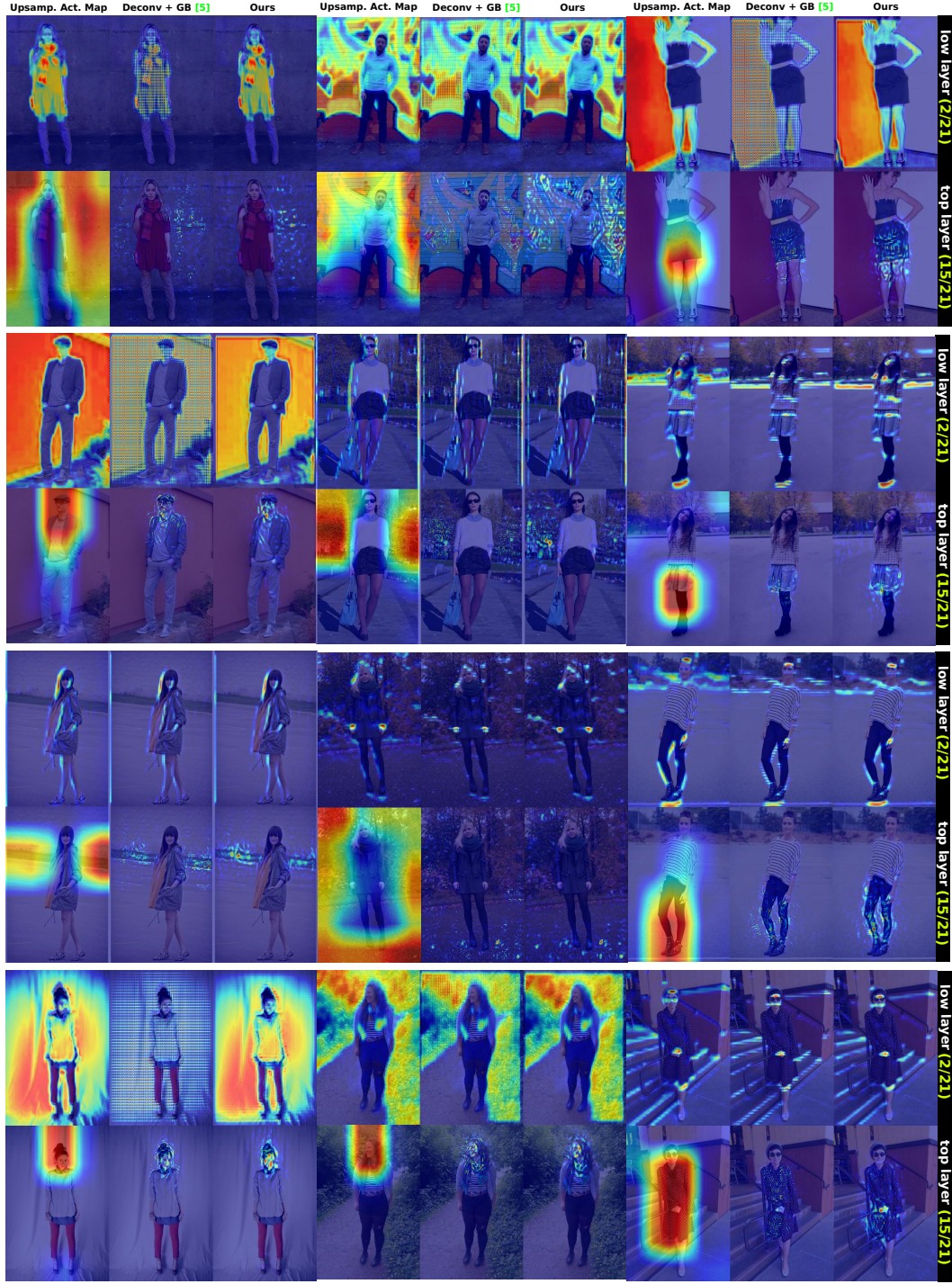

Figure 26: **Visual quality comparison** for visualizations generated from the **Fashion114k** dataset Simo-Serra et al. (2015). Note how our heatmaps attenuate the grid-like artifacts introduced by deconvnet-based methods at lower layers. Likewise, our method is able to produce a more detailed visual feedback than upsampled activation maps.

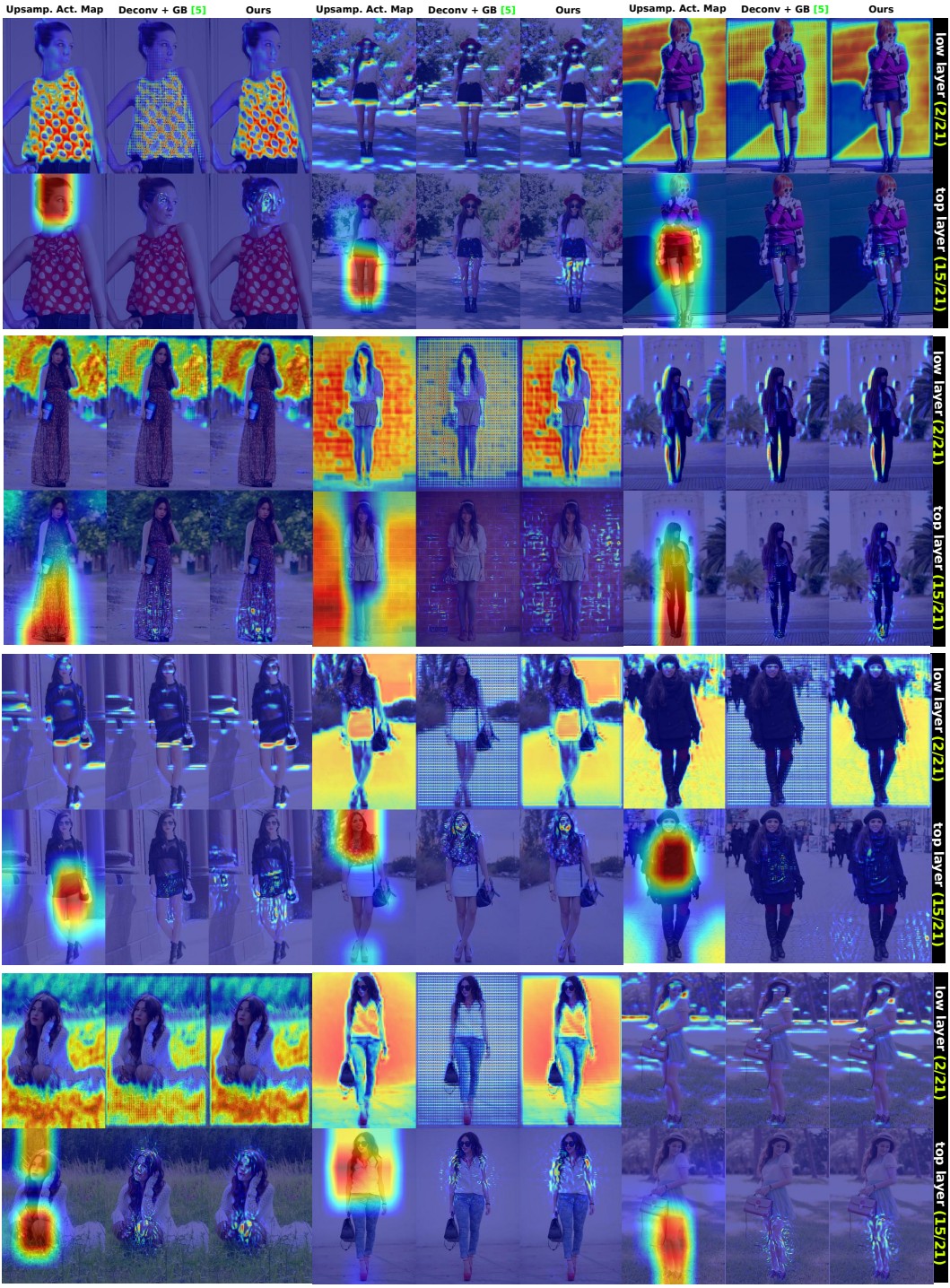

Figure 27: **Visual quality comparison** for visualizations generated from the **Fashion114k** dataset Simo-Serra et al. (2015). Note how our heatmaps attenuate the grid-like artifacts introduced by deconvnet-based methods at lower layers. Likewise, our method is able to produce a more detailed visual feedback than upsampled activation maps.

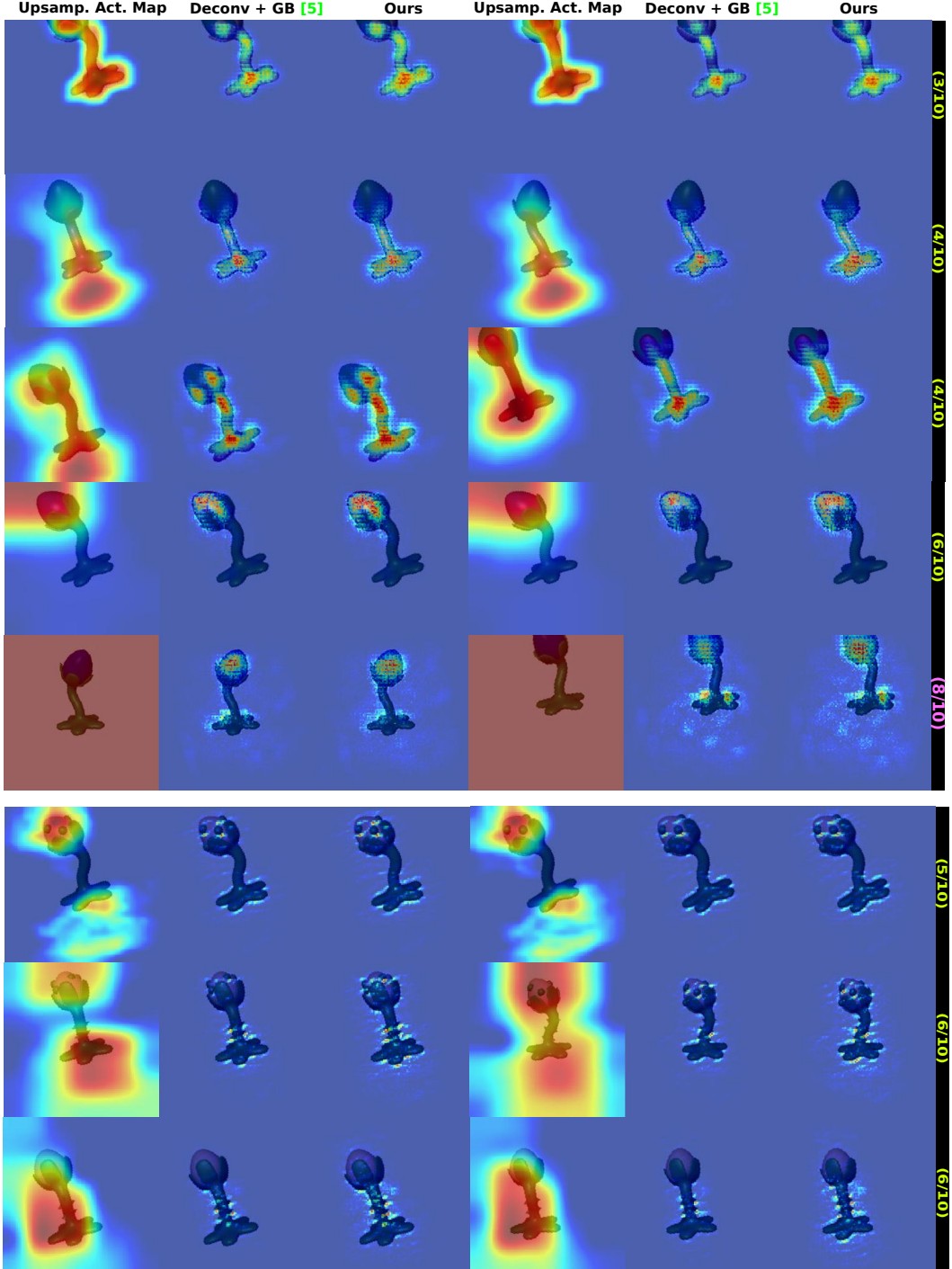

Figure 28: **Visual quality comparison** for visualizations generated from the **an8Flower** dataset. The ground truth mask of the first three rows is stem, middle two rows are flowers while the last three rows are balls (on flower), thorns (on stem) and both. Note how our heatmaps have a higher coverage and stronger response in the ground truth mask area. Likewise, our method is able to produce a more detailed visual feedback than upsampled activation maps.

