# OpenReview forum: "Visual Explanation by Interpretation: Improving Visual Feedback Capabilities of Deep Neural Networks"
_ICLR.cc/2019/Conference_

### Official Review · AnonReviewer3 · 2018-11-04
**A good paper with interesting contributions that requires improved motivations and some clarifications**

**Rating:** 5
**Confidence:** 3

**Review:**

Summary: the paper proposes a method for Deep Neural Networks (DNN) that identifies automatically relevant features of the set of the classes, enriching the predictions made with the visual features that contributed to that class, supporting, thus, interpretation (understanding what the model has learned) and explanation (justification of the predictions/classifications made by the model). This scheme does not rely on additional annotations, like earlier techniques do.

The contributions of this paper are relevant to, I would say, a large segment of the AI community, since interpretability and explainability of AI (XAI) is the focus of many current works in the area, and there are still many unresolved issues. I consider this paper suitable for ICLR 2019, in particular, it fits the call for papers topic “visualization or interpretation of learned representations”.

The authors also present a new dataset (am8Flower) that can be used by the community for future evaluations of explanation methods for DNN. From my point of view, this is a significant contribution, since there is a lack of datasets that can be used for evaluation.

The authors motivate properly the need for this research/study, addressing the main weakness of the two more common strategies for interpreting DNN, (1) manually inspecting visualizations of every single filter or (2) comparing the internal activations produced by a given model w.r.t. a dataset with pixel-wise annotations of possibly relevant concepts.

I would encourage the authors to write the limitations and weakness of their proposal w.r.t. similar approaches they reviewed. I am aware that the space is limited, but in p.8, section 4.3, when Table 1 is introduced and the authors confirm that their proposal has higher IoU than other methods, the authors could explain, in brief, what are the weaknesses of their method w.r.t. the other approaches analyzed.

Another clarification concerns the initialization of input parameters, such as sparsity; e.g., p.6 sparsity is initialized with 10 for all datasets, why? How has this value been selected and how sensitive is the performance regarding variations of this value?

Once again, I know that the space is limited, but I would like to be able to see some of the figures better (since this is an essential part of the paper). The additional material complements very well the paper and shows larger figures, but I think that the paper itself should be self-sufficient, and figures like Fig. 5 should be enlarged so it is easier to see some details.

Just a concern or something that I quite did not understand about one of the arguments the authors use to justify the evaluation carried out: the authors claim that they want to avoid the subjectivity introduced by humans (citing Gonzalez-Garcia et al. 2017), and prefer to avoid user studies, presenting a more objective approach in their evaluation. Ok, but then, the analysis presented in, for example, page 7, is based mainly in their interpretation of the results, a qualitative analysis of the images (we can see fur patterns, this and that, etc.). So aren’t they interpreting the results obtained as users? So after all, aren’t the visual explanations and feedback intended for users? Why should we claim that we want to avoid the subjectivity introduced by humans in the evaluation when the method proposed here is actually going to be used by users –with their inherent subjectivity? I do not mean that the evaluation carried out is not interesting per se, but it could be motivated differently, or it could be complemented later on with future user studies (that would make an interesting addition to the paper). Moreover, I also wonder whom the authors see as intended users for the proposed scheme.

Small comments:
P.1 “useful insights on the internal representations”  insights into the internal representations.
P. 2: space needed in “back-propagation methods.Third,”
P. 3: Remove “s” in verb (plural authors): “Similarly, Bach et al. (2015) decomposes the classification”  decompose or decomposed
P.3: n needed “Chattopadhyay et al. (2018) exteded”  extended
P.3: “This saliency-based protocol assume that”  protocol assumes
P.3: “highlighted by the the explanations”  remove one “the”
P. 5: “space. As as result we get”  remove one “as”
P. 5: “and compensate this change”  compensate for this change
P. 6: “In this experiment we verify”  In this experiment, we verify
P. 6: “To this end, given a set of identified features we”  To this end, given a set of identified features, we
P. 6: “Note that the OnlyConv method, makes the assumption”  remove “,” after method
P. 7: “In order to get a qualitative insight of the type of”  insight into the
P. 7: I would write siamese and persian cat with capital “S” and “P” (Siamese, Persian)
P. 7: others/ upper “Some focus on legs, covered and uncovered, while other focus on the upped body part.”  while others focus on the upper body part
P. 7: “These visualizations answers the question”  answer
P. 7:  “In this section we assess”  In this section, we
P. 7: Plural “We show these visualization for different”  these visualizations
P. 7: In “Here our method reaches a mean difference on prediction confidence”  difference in prediction …
P. 7: “This suggest that our method is able”  This suggests that
P. 8: state-of-the-art
P. 8: “has higher mean IoU”  has a higher mean IoU
Whole document: when using “i.e.” add “,” after: i.e.,

References: Some of the references in the list have very little information to be able to find it/proper academic citation, e.g. , Yosinski et al. 2015; Vedaldi and Lenc, 2015:

Jason Yosinski, Jeff Clune, Anh Mai Nguyen, Thomas J. Fuchs, and Hod Lipson. Understanding neural networks through deep visualization. 2015.

A. Vedaldi and K. Lenc. Matconvnet: Convolutional neural networks for matlab. In MM, 2015.

Ref Doersch et al.: What makes paris look like paris?  Paris

---

> ### Author Response · Authors · 2018-11-19
> **(2|2) : RE: A good paper with interesting contributions that requires improved motivations and some clarifications**
>
>
> We consider as potential users of our method,
>
> - For dataset debugging
> Our method can assist other researchers on verifying whether a top-performing model has indeed learned a general representation from the data or this top performance is caused by any bias on the data itself, e.g. dataset bias.
>
> - For accountability
> It can serve students, researchers and other individuals working with deep models identify relatively not apparent causes for the high performance of a given model. For instance, our method has provided a set of good insights of why the models trained in Wang et al., WACV'18 have such a good performance when compared with human subjects.
>
> - To enforce fairness
> Very related to the previous case, individuals tasked with assessing the "fairness" of models making decisions about other individuals (e.g Gender Shades, Buolamwini et al., PMLR'18). Our method can help to verify whether these models have any bias related to genre, ethnicity, etc.
>
> - Alternatively, our visual explanations can serve to indicate regions of interest that can serve as input to other automatic systems/methods aiming at distilling information from the model being explained with the goal of producing lighter models.
>
>
> We thank the reviewer for the time invested in providing the detailed feedback at the end of the review (small comments and references). Likewise, we will invest time to meticulously integrate this feedback on a revised version of our manuscript.
>
> We have revised the manuscript in order to integrate the provided feedback.
> (New content is coloured in green)

---

> ### Author Response · Authors · 2018-11-19
> **(1|2) : A good paper with interesting contributions that requires improved motivations and some clarifications**
>
> Thanks for the feedback.
>
> We appreciate the reviewer recognizes the relevance that our work can have in the field in general.
> We agree with the reviewer on the significance of the contribution given by the proposed an8Flower dataset.
> Although very recently few works (Adebayo et al.,NIPS'18, Nie et al., ICML'18) have proposed means to assess the sanity/reliability of visual explanation methods, no method has been proposed to objectively evaluate the generated explanations themselves.
> Moreover, the proposed an8Flower dataset can be further extended to evaluate different settings of interest, e.g. occurrence of distracting objects, object classes driven by contextual information, fine-grained differences classes, etc., and can be used as a sanity check itself to verify whether a proposed explanation method can accurately explain a specific setting of interest.
>
> Regarding the suggestion of providing a discussion covering the limitations/weaknesses of the proposed method w.r.t. similar compared methods, e.g. those from Table 1.
> Indeed, there are space limitations in place, as was pointed out by the reviewer.
> However, this is a good suggestion and we believe that adding such discussion would provide further insights on the proposed method, and strengthen the manuscript at the same time.
> Here is a summary of limitations that our method has w.r.t. similar compared methods:
> - Our method requires an additional process, i.e. feature selection via u-lasso, at training time (Sec. 3.1).
> - There is the need to define an additional parameter, i.e \mu, for the feature selection process (Sec. 3.1).
>
> Regarding the sparsity parameter (\mu) used for the feature selection process (u-lasso): increasing the sparsity value \mu in the u-lasso formulation will increase the number of selected features.
> This will allow to the selected filters to focus on more specific/specialized features that can help to handle better outlier/rare instances of the classes of interest. Please see, Sec.8 and Fig.9 from the supplementary material for an extra analysis on the effect that the \mu value has on the capability of the selected features to serve as indicators of the classes of interest.
> We decided to start from a relatively low value, i.e. \mu=10, in order to focus on a small set of relevant features that can generalize to the classes of interest while, at the same time, keeping the u-lasso optimization with a low computational cost.
>
> Regarding the size of the figures (Fig.5 especially), we totally agree with the reviewer that figures like Fig. 5 should be enlarged so it is easier to see some details. Despite the space limitations, we are aware that the 8-page length for the manuscript (content only) is not strict, and that authors are allowed to go up to 10 pages. Having said this, if reviewers and ACs agree on the need for larger figures, we would like to cross the 8-page length and include larger versions of some of the figures that are currently too small to visualize details.
>
> Regarding the question of whether human inspection (or user studies) are necessary for model interpretation/explanation.
> We agree with the observation made by the reviewer regarding the fact that our method still requires some level of human intervention. Furthermore, we agree that since the proposed method is meant to be used by users, an indication of how "understandable" an explanation is for end users is required. Having said this, the main goal of our method is to reduce the load on the user side which can introduce bias and noise. By reducing (and separating) the number of visualizations (i.e. number of features in the explanation visualizations and the relevant set of features learned by the model [interpretation]) to be inspected, we aim at reducing exhaustive inspections that are used in previous works to achieve model interpretation/explanation.
> We admit that in our manuscript, the need for human inspection is understated. Moreover, we agree that our objective evaluation should be complemented with relatively simpler user studies in order to ensure that the produced explanations are meaningful to the individuals they aim to serve. We will update the motivation behind our method in order to emphasize further the need of reduced used inspection and the complementary between our evaluation and user studies.

---

### Official Review · AnonReviewer1 · 2018-11-05
**Some issues**

**Rating:** 4
**Confidence:** 5

**Review:**

In this paper, the authors proposed a novel scheme to interpret deep neural networks’ prediction by identifying the most important neurons/activations for each category using a Lasso algorithm.

Firstly, the authors produce a 1-dimensional descriptor for each filter in each convolutional layer for each image. Then these descriptors are concatenated as a new feature vector for this image. A feature selection algorithm (u-Lasso) is then trained to minimize the classification loss between the prediction from the new feature vector and the original prediction from DNN (formula (1)). Finally, the importance of each filter is identified by the weights of the lasso for each category.

The authors also improved the visual feedback quality over the deconvolution+guided back-propagation methods, and release a new synthetic dataset for benchmarking model explanation.

The paper is well-written, however, I have several concerns about this paper:

1.      How to verify the importance of the identified relevant features is a problem. In the experiments, the authors removed features in the network by setting their corresponding layer/filter to zero. The authors only compared their method with randomly removing features. And in Fig 4, the differences seem small for ImageNet. The results are not convincing enough to me. It is a bit baffling randomly removing features did almost as well as the proposed approach.

2.      I don't think one should get away with only showing some results from the synthetic dataset without showing any quantitative results on any real datasets. I like the idea of having a synthetic dataset where all the parameters are controllable. However in this case it is very simple and maybe lacking enough distracting features that can really test the capability of the algorithm. I would believe quantitative results on a realistic dataset are still necessary for the pubilcation of this paper.

3.      Recently several papers pointed out some significant issues in Guided BP,

Xie et al. A Theoretical Explanation for Perplexing Behaviors of Backpropagation-based Visualizations. ICML 2018
Adebayo et al. Sanity Checks for Saliency Maps. NIPS 2018
Kindermans et al. The (Un)reliability of saliency methods. NIPS workshop 2017

can the authors comment on that? Based on those papers I don't seem to think Guided BP is actually doing anything that is relevant to the classification, but is just finding prominent gradients. This, unfortunately would lead to reasonably good behavior on the synthetic dataset created by the authors.

---

> ### Author Response · Authors · 2018-11-19
> **(2|2) : RE: Some issues**
>
>
> Regarding (3), thanks for the pointers towards those works. Indeed there are some interesting insights there that we can address from the perspective of our method.
>
> As pointed by the reviewer, our method identified important features for each of the classes modeled by the network using a u-Lasso optimization. Then, at test time, we explain the class predicted by the model by, first, looking for the response (on the test image) of the subset of features identified as relevant for such class, and then, generating heatmaps highlighting the top responding features via our variant of DeconvNet with Guided Backpropagation. Each of these feature visualizations is generated by using the DeconvNet with Guided Backpropagation method to highlight the image regions that produce the activations observed for the relevant features.
> As such our method is composed by two main components: a) the feature selection component, and b) the visualization component. At test time, these two components are linked by the class predicted by the model.
>
> Kindermans et al., NIPS'17 (ws), propose a shift-test in which the explanations produced by a image-model pair should match that of its shifted counterpart.
> In our case, relevant features are identified by applying the u-Lasso optimization on the internal activations. If these activations remain constant (as enforced by the shift-test) no difference in the selected features (filter/layers) is to be expected.
>
> Nie et al.,ICML'18 and Adebayo et al., NIPS'18 suggest that explanations from DeconvNet and Guided-Backpropagation methods are not determined by the predicted class, but by the filters of the first layer and the edge-like structures in the input images.
> Regarding the question of whether this is the reason why our method has good performance on the proposed dataset, it can be noted that the explanations generated by our method in the proposed dataset go beyond regions with prominent gradients (edge-like regions). In fact, in classes where color is a discriminative feature uniform regions are highlighted.
> Moreover, in our method, we use DeconvNet with Guided-Backpropagation as means to highlight the image regions that justify the identified relevant features, not the predicted classes themselves.
>
> Regardless of the observations made in the referred works, the adopted DeconvNet+GBP method is just the means we use for visualization. This visualization method does not influence the way in which relevant features are selected but in the way they are visualized. So if a better, more robust/principled, visualization method is proposed in the literature it can be integrated into an upgraded version of our method. We don't see the current visualization mechanism as a major weakness but as a point that can be improved as new insights related to visualization of internal features of DNNs are obtained by the community.
>
> We thank the reviewer for motivating the discussion in the direction of these aspects. We believe the discussion above helps to get an insight on strengths and potential points for improvement of the proposed method. If the other reviewers and ACs agree, we would like to add the discussion above in a revised version of the manuscript. Additionally, depending on the time (and space) constrains, we will try to add some of the tests presented in the referred papers from above in a revised version of our submission.
>
> We have revised the manuscript in order to integrate the provided feedback.
> (New content is coloured in green)

---

> ### Author Response · Authors · 2018-11-19
> **(1|2) : RE : Some issues**
>
> Thanks for the feedback.
>
> Regarding (1), the ablation of features labeled as "random" refers to settings where features were removed by setting to zero the response of randomly selected filters from layers that were indicated to host important features by the u-lasso optimization.
> As such, these features (filter responses) are not 100% random per se. To verify this aspect, we have conducted an experiment on the full imageNet dataset where we ablated completely randomly selected features  (i.e  both layers and filter locations). We computed the mean performance after 5 runs and obtained a classification accuracy of 0.33, which is  10% higher than that when the selected relevant features are ablated (0.23).
> In addition, different from the other datasets tested with a VGG-based method, the setting of the full imageNet dataset has the highest ratio between classes of interest and features. At this higher ratio, features internally modeled by the network are more likely to be shared between the classes. As such, ablating one feature may have a side effect on another class as well.
>
>
> Regarding (2), we respectfully disagree. Our synthetic dataset may look simple and artificial, but that's on purpose to make it clear beyond discussion what elements are crucial to explain a decision. To the best of our knowledge there isn't any realistic dataset with such annotation and in fact, we have no idea how one would go about to create one. Nor is there any other unbiased quantitative evaluation setup using realistic data, as far as we know. For instance, using semantic labels as done in (Zhang et al, CVPR'18 / arXiv:1710.00935 ignores the validity of any context cues that fall outside of the object boundaries. In our synthetic dataset, the regions to be highlighted are controlled by design, therefore providing an objective means of evaluation. If however the reviewer can point us towards a realistic dataset which such level of annotation, we would be happy to try it out, to further strengthen our manuscript.
>
> Regarding the presence of distracting features mentioned in (2), we are conducting experiments on the classification task of Pascal VOC'07. In this dataset there are several distracting instances/objects per image. Initial results show that despite the presence of these distracting objects/elements, our method is able to highlight image regions related to the prediction made by the model. If requested by the reviewers, we will revise the manuscript in order to include some of these new results.
> In addition, adding distracting objects/features could be an interesting way to extend our current synthetic dataset. We will work towards having an additional variant of our dataset that includes distracting elements for the moment of its official release.

---

### Official Review · AnonReviewer2 · 2018-11-06
**A great contribution to the visual explanation literature!**

**Rating:** 8
**Confidence:** 4

**Review:**

Pros:

This paper
 - Proposes a method for producing visual explanations for deep neural network outputs,
 - Improves quality of the guided backprop approach for strided layers by converting stride 2 layers to stride 1 and resampling inputs (improving on a longstanding difficulty with such approaches),
 - Shows fairly rigorous experimentation demonstrating the applicability and properties of the proposed approach, and
 - Releases a new synthetic dataset and benchmark for visual explanation methods.

Although producing visual explanations is a task fraught with difficulty for many reasons, including that explanations for complex decisions may not necessarily be communicable via one or a small number of saliency maps over the image pixels, this paper strives valiantly in this admittedly difficult direction.

The experimentation is fairly rigorous, which is a welcome departure from and improvement on the norm for this type of paper. I hope such more quantitative evaluation will become more common in papers evaluating visual explanations.

Cons:

What about features that are very important but not linearly predictive on their own? This approach (and many others) would not work in that case; recognizing this, extending the an8Flower dataset to include such images and labels may be motivating for the field. For example, flowers where the class is determined not by a specific single color or feature (thorns or spots) but by the combination. In these cases, it’s not clear what the right answer would even be in the form of a saliency map, so the first task for researchers would be to determine in what format the answer should even be provided! So: less a benchmark than a motivating open question.


Smaller notes:

I found the presentation of the stride 1 resampling approach a little confusing. When performing the backward pass through the network from, say, layer 20, is the approach followed at every stride 2 layer on the way back? If so, I don’t think I saw this mentioned. If not, wouldn’t artifacts be introduced and compounded at any stride 2 layer during the backward pass?


====== Update 12/12/18 ======

Thanks for your notes in reply. I'll just add that if the dataset can be extended to slightly greater complexity either for this version or for submission to a subsequent venue, it would be impactful. Simple extensions could include scenes with multiple flowers and classes where the explanatory factor is tricker to uncover. For example, a dataset could be created with scenes of three flowers: two of one color and one of another color, with the class determined by the color of the lone flower. The correct explanation (the color of the lone flower) is still clear, and it would be great to see if the proposed LASSO approach (or a future approach) could correctly identify those pixels.

---

> ### Author Response · Authors · 2018-11-19
> **RE: A great contribution to the visual explanation literature!**
>
>
> We thank you for the motivating feedback. As you mentioned visual explanation is a task fraught with difficulty for many reasons, and indeed here we try to push efforts addressing this task, with important neuron selection, better visualization and especially new complementary objective evaluation protocols.
>
> Regarding the comment on the features that are very important but not linearly predictive: we tried to cover that scenario to some extent with the an8flower-double-12c variant of our dataset (please see Fig. 10 from the supplementary material). There the classes are not just defined by the color, but by the part/location where those colors are applied. Yet, this is just one scenario; there are many others that might be interesting to investigate. In its current form, an8Flower is just an initial step towards more objective evaluation. Taking into account the feedback from reviewers and from the community, we hope to turn it into a fully developed benchmark for visual explanations.
>
> Regarding the comment "...so the first task for researchers would be to determine in what format the answer should even be provided!". Indeed that is a very good point and an interesting research question. We hope to be able to tackle such questions in future work.
>
> Regarding the smaller note on the stride 2 resampling approach: yes, you are right, this adjustment is applied to every layer during the backward pass. Otherwise, as accurately noted, artifacts produced at top layers would propagate towards the lower ones.
> We thank the reviewer for pointing this out. We will revise the manuscript to make sure this aspect is clear.
>
> We have revised the manuscript in order to integrate the provided feedback.
> (New content is coloured in green)

---

### Meta-Review · Area_Chair1 · 2018-12-17

**Confidence:** 4
**Recommendation:** Accept (Poster)

**Metareview:**

This was a difficult decision to converge to. R2 strongly champions this work, R1 is strongly critical, and R3 did not participate in the discussions (or take a stand). On the one hand, the AC can sympathize with R1's concerns -- insights developed on synthetic datasets may fail to generalize and fundamentally, the burden is not on a reviewer to be able to provide to authors a realistic dataset for the paper to experiment on. Having said that, a carefully constructed synthetic dataset is often *exactly* what the community needs as the first step to studying a difficult problem. Moreover, it is better for a proceeding to include works that generate vigorous discussions than the routine bland incremental works that typically dominate. Welcome to ICLR19.